# Kynurenine-3-monooxygenase (KMO) broadly inhibits viral infections via triggering NMDAR/Ca$^{2+}$ influx and CaMKII/ IRF3-mediated IFN-β production

Jin Zhao[1,2], Jiaoshan Chen[1,2], Congcong Wang[1,2], Yajie Liu[1,2], Minchao Li[1,2], Yanjun Li[1,2], Ruiting Li[1,2], Zirong Han[1,2], Junjian Wang[3], Ling Chen[4], Yuelong Shu[1,2], Genhong Cheng[5]*, Caijun Sun[1,2]*

1 School of Public Health (Shenzhen), Shenzhen Campus of Sun Yat-sen University, Shenzhen, China, 2 Key Laboratory of Tropical Disease Control (Sun Yat-sen university), Ministry of Education, Guangzhou, China, 3 Guangdong Provincial Key Laboratory of New Drug Design and Evaluation, School of Pharmaceutical Sciences, Sun Yat-sen University, Guangzhou, China, 4 State Key Laboratory of Respiratory Disease, Guangzhou Institutes of Biomedicine and Health (GIBH), Chinese Academy of Sciences, Guangzhou, China, 5 Department of Microbiology, Immunology and Molecular Genetics, University of California, Los Angeles, California, United States of America

* gcheng@mednet.ucla.edu (GC); suncaijun@mail.sysu.edu.cn (CS)

**Data Availability Statement:** All relevant data are within the manuscript and its Supporting Information files.

## Abstract

Tryptophan (Trp) metabolism through the kynurenine pathway (KP) is well known to play a critical function in cancer, autoimmune and neurodegenerative diseases. However, its role in host-pathogen interactions has not been characterized yet. Herein, we identified that kynurenine-3-monooxygenase (KMO), a key rate-limiting enzyme in the KP, and quinolinic acid (QUIN), a key enzymatic product of KMO enzyme, exerted a novel antiviral function against a broad range of viruses. Mechanistically, QUIN induced the production of type I interferon (IFN-I) via activating the N-methyl-d-aspartate receptor (NMDAR) and Ca$^{2+}$ influx to activate Calcium/calmodulin-dependent protein kinase II (CaMKII)/interferon regulatory factor 3 (IRF3). Importantly, QUIN treatment effectively inhibited viral infections and alleviated disease progression in mice. Furthermore, *kmo$^{-/-}$* mice were vulnerable to pathogenic viral challenge with severe clinical symptoms. Collectively, our results demonstrated that KMO and its enzymatic product QUIN were potential therapeutics against emerging pathogenic viruses.

## Author summary

The outbreaks of emerging infectious diseases have become a severe challenge worldwide, and therefore it is a public health priority to explore novel broad-spectrum antiviral agents with various mechanisms. This study reported that kynurenine-3-monooxygenase (KMO), a key rate-limiting enzyme during tryptophan metabolism, showed promise as a novel broad-spectrum antiviral factor against emerging pathogenic viruses. We further found that quinolinic acid (QUIN), an enzymatic product of KMO, could also act as a

**Funding:** This work was supported by the National Key R&D Program of China (2021YFC2300103; Recipient C.S.), the National Natural Science Foundation of China (81971927; Recipient C.S.), and the Science and Technology Planning Project of Shenzhen City (20190804095916056, JSGG20200225152008136, JCYJ20200109142601702; Recipient C.S.). The funders had no role in study design, data collection and analysis, decision to publish, or preparation of the manuscript.

**Competing interests:** The authors have declared that no competing interests exist.

novel broad-spectrum antiviral agent. We then systematically studied the underlying mechanisms and broadly antiviral function of KMO and QUIN *in vitro* and *in vivo*. Our data highlight the importance of exploring novel antiviral targets from the key enzymes and their metabolites in tryptophan metabolism.

## Introduction

Frequent outbreaks of emerging infectious diseases, including SARS-CoV-2, avian influenza H5N8, Zika virus (ZIKV), and Ebola virus (EBOV), have become a serious threat to global public health [1–6], and an urgent need in clinical practice is to explore the efficient and broad-spectrum antiviral agents against various viral infections. Increasing studies indicated the potential cross-talk between the immune responses to viral infections and the cellular metabolism in host cells [7,8]. In addition, viral infections may hijack the immune system and the metabolic system and thus cause metabolic disorders, including abnormal lipid metabolism, cardiovascular diseases, and neurological diseases [9–11]. As a result, in-depth investigation of the relationship between immune response and cellular metabolism may reveal novel targets to develop antiviral agents.

L-Tryptophan (L-Try) is a kind of essential amino acid for the human body, and its metabolites by the kynurenine pathway (KP) have critical functions in inflammation and immune homeostasis, thereby playing a key role in autoimmune diseases, tumorigenesis, and neurodegenerative diseases [12–15]. Recent studies also suggested that KP metabolites might play a regulatory role in host-pathogen interactions [16–18]. For example, the transcription level of indoleamine 2,3-dioxygenase 1 (IDO1), one metabolite of tryptophan, was significantly upregulated in response to influenza infection in mouse lungs and human primary macrophages [19,20]. Moreover, kynurenine biosynthesis in macrophages was increased in response to stimulation by the herpes simplex virus (HSV) as well as bacterial lipopolysaccharides [21]. In addition, another metabolite of KP-picolinic acid effectively induced the apoptosis of human immunodeficiency virus (HIV) or HSV-infected cells [22]. Thus, it is important to investigate further the host-pathogen interactions through the KP metabolites to explore the novel anti-infection targets.

Kynurenine-3-monooxygenase (KMO), a key rate-limiting enzyme during L-Try metabolism, belongs to the family of oxidoreductases. KMO can catalyze the conversion of L-kynurenine to 3-hydroxy-L-kynurenine and also regulate the balance between kynurenic acid (KA) and quinolinic acid (QUIN) [15]. So far, the role of KMO in innate immunity response to viral infections has not been reported yet. In the present study, we identified for the first time that KMO and its enzymatic product QUIN could act as a novel broad-spectrum antiviral factor against emerging pathogenic viruses.

## Result

### KMO is an interferon-dependent gene

To identify the potential genes with antiviral activity, we used a recombinant HSV-GFP-Luc reporter virus to screen the 288 ISGs induced by IFN-α and IFN-γ as we previously reported [23]. Besides the well-known antiviral genes including IRF1 [24], CH25H [25], IFITM2 [26], TAP1 [27], LY6E [28], we also identified some novel genes including KMO, SERPINA5, SLFN12, GPR146, and LIPG, which are uncharacterized with antiviral activity previously (Fig 1A). Our

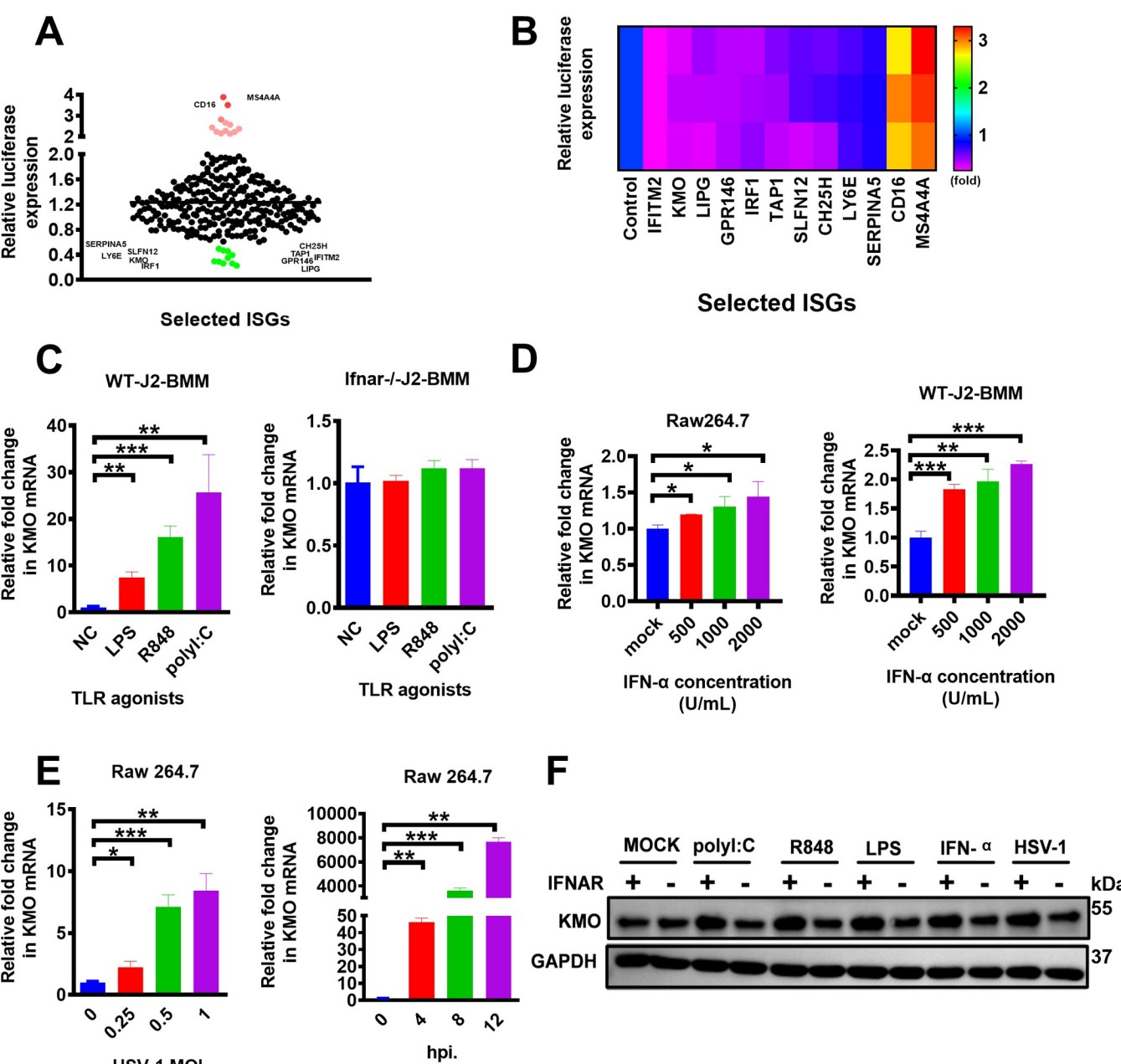

**Fig 1. The expression of KMO is interferon-inducible.** (A,B) 288 genes were screened by a luciferase-based analysis. Each dot represents one gene, and its effect on HSV-1 infection was normalized to that in pVAX-GFP transfected cells (negative control). Genes that inhibit HSV-1 infection by more than 50% were marked and shown in green. Genes that enhance HSV-1 infection by more than 2 fold were marked and shown in red. (C) Wild type bone marrow-derived macrophage (WT-J2-BMM) cells or interferon α receptor-deficient (*Ifnar^-/-*)-J2-BMM cells were stimulated for 8 h with different TLR agonists, including LPS (1 µg/mL), R848 (100 nM), poly(I:C) (25 µg/mL), and then the expression level of *kmo* gene was measured by RT-qPCR. (D) Raw264.7 cells and J2-BMM cells were stimulated with IFN-α in different concentrations (500 U/mL, 1000 U/mL, and 2000 U /mL) for 8 h, and the expression level of the *kmo* gene was measured by RT-qPCR. (E) Raw264.7 cells were infected with indicated MOI of HSV-1 for 24 h or infected with various times (hours post-infection, hpi.) at 0.25 MOI of HSV-1. Then, the expression level of the *kmo* gene was measured by RT-qPCR. (F) J2-BMM cells and *Ifnar^-/—*J2-BMM cells were stimulated with LPS (1 µg/mL), poly(I:C) (25 µg/mL), R848 (100 nM), IFN-α (2000 U/mL), or HSV-1 at MOI of 0.25 for 24 h, respectively, and then the expression level of KMO protein was measured by Western Blotting Analysis. GAPDH was used as intern control. The final data are presented as the mean ± SD of at least triplicate experiments. MOI: multiplicity of infection. $^*P < 0.05$, $^{**}P < 0.01$, $^{***}P < 0.001$.

further analysis demonstrated that IFITM2, KMO, LIPG genes exerted the most obvious inhibition against viral infections in this study (Fig 1B).

Next, we sought to characterize the antiviral function of the KMO gene since it was previously well-described as a key rate-limiting enzyme in KP-mediated L-Try metabolism [29]. Our results showed that KMO expression was significantly upregulated in response to the stimulation of different toll-like receptors (TLRs) agonists, such as LPS, poly(I:C), R848 (Fig 1C). However, the up-regulation of KMO expression was abrogated in the IFN-I receptor (IFNAR) deficient (*Ifnar^-/-*)-J2-BMMs (Fig 1C). In addition, in a dose-dependent manner, KMO expression was significantly induced by IFN-α stimulation in Raw264.7 and WT-J2-BMM cells (Fig 1D). We also found that KMO expression was significantly elevated by HSV-1 infection in both dose-dependent and time-dependent in Raw264.7 cells (Fig 1E). The Western Blotting analysis further confirmed that KMO protein was increased by the agonists mentioned above in J2-BMM cells while abolished in the (*Ifna^-/-*)-J2-BMM cells (Fig 1F). Altogether, the KMO expression is IFN-dependent.

## The antiviral activity of KMO

Next, we investigated the potential antiviral activity of KMO. Using the CH25H gene as a positive control [30], we demonstrated that KMO over-expression significantly suppressed the replication of HSV-1 by quantitative PCR analysis (Fig 2A), and this inhibition was further confirmed by Western Blotting analysis (Fig 2B) and plaque assay (Fig 2C). To further validate the antiviral function of KMO, the KMO expression was effectively inhibited using small interfering RNA (siRNA) (Fig 2D) and short hairpin RNA (shRNA) (Fig 2E), respectively. Results demonstrated that HSV-1 replication was significantly enhanced in KMO-knockdown cells compared to wild-type cells (Figs 2D and 3E). Furthermore, we generated the *kmo^-/-* cell line using the CRISPR/Cas9 system (Fig 2F) and found a significantly enhanced HSV-1 replication in *kmo^-/-* cells (Fig 2G) compared to wild-type (WT) cells. Moreover, the replenishment of the *kmo* gene could effectively rescue the antiviral activity in *kmo^-/-* cells (Fig 2H). Altogether, our studies with both loss-of-function and gain-of-function experiments validated the antiviral function of the *kmo* gene.

## The antiviral effects of KMO through its enzymatic product QUIN

We also explored whether the antiviral function of KMO is dependent on its enzyme activity. Previous studies showed that the residues Tyr 99, Tyr 194, and Glu 366 were critical to the enzymatic activity of KMO [31,32], and we subsequently generated a series of *kmo* mutants (KMO-M1-3) by site-directed mutagenesis (Fig 2I). Results showed that the overexpression of these KMO enzyme-dead mutants had no significant inhibition effect on HSV-1 infection compared to that of wild-type KMO (Fig 2J), implying that the enzymatic activity of KMO is required for its antiviral function.

We subsequently assessed whether the enzymatic products in the supernatants of KMO-treated cell cultures exerted the viral inhibition effect. As expected, the supernatants from KMO-treated cells significantly inhibited HSV-1 infection in Vero cells and 293T cells (Fig 2K), and Western Blotting analysis further confirmed this observation (Fig 2L), suggesting that KMO exerted antiviral function by producing potential soluble antiviral factors. KMO is a well-known key enzyme in the KP to produce many kinds of metabolites [29–33], including quinolinic acid (QUIN), kynurenic acid, xanthic acid. Among them, we identified that QUIN had a significant antiviral effect in a dose-dependent manner (Fig 2M). Moreover, QUIN also effectively rescues the antiviral activity in *kmo^-/-* cells (S1 Fig). In addition, we investigated the

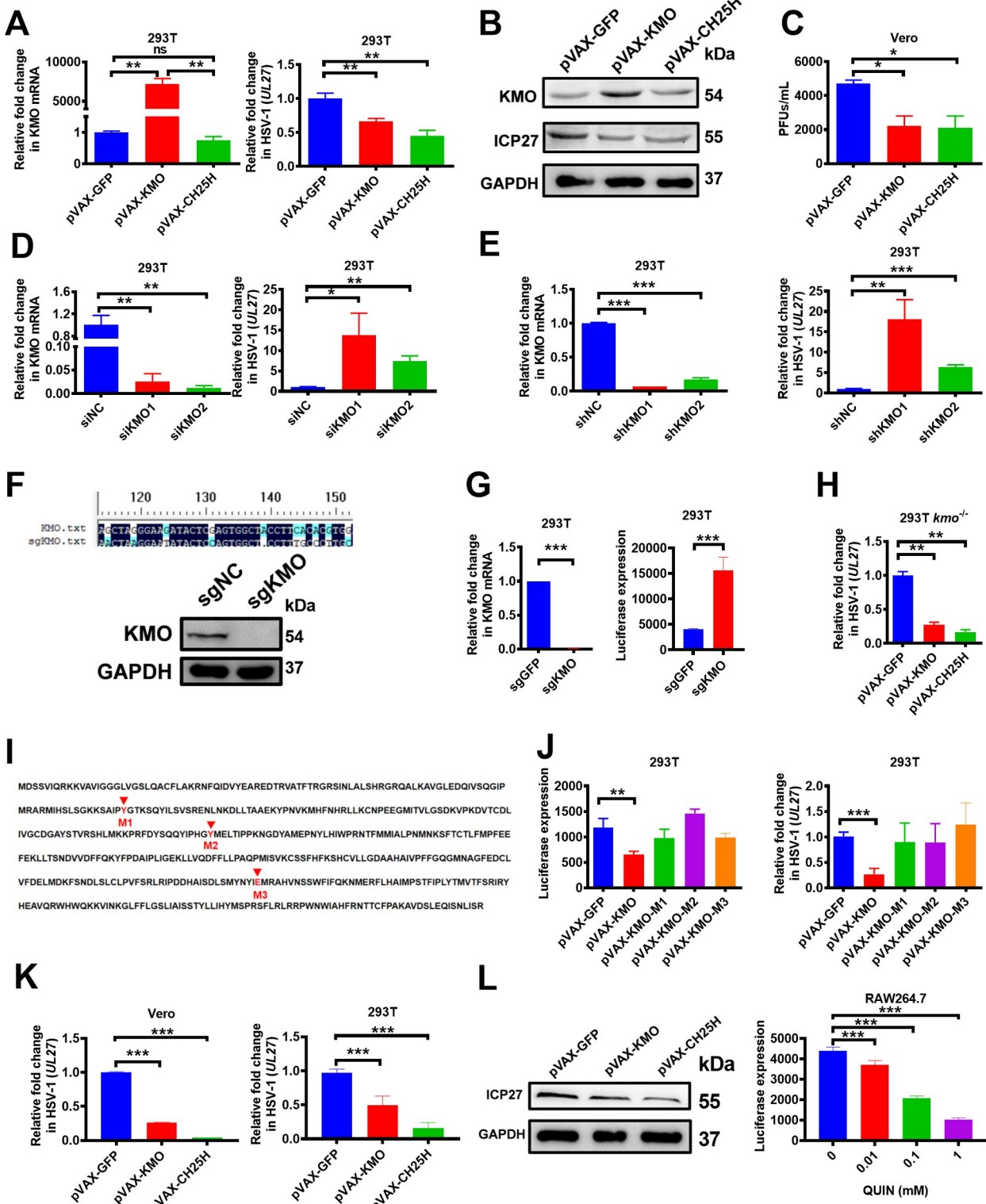

**Fig 2. The antiviral activity of KMO through its enzymatic product QUIN.** (A) 293T cells were transfected with plasmids (1 μg/mL) expressing GFP, KMO, CH25H for 24 h respectively, followed by HSV-1 infection for 8 h at MOI of 0.25. Then the expression of KMO and HSV-1 UL27 were measured by RT-qPCR. (B) 293T cells were transfected with plasmids (1 μg/mL) expressing GFP, KMO, CH25H for 24 h respectively, followed by HSV-1 infection for 8 h at MOI of 0.25. Then the expression of KMO and HSV-1 ICP27 protein were measured by Western Blotting Analysis. (C) 293T cells were transfected with plasmids (1 μg/mL) expressing GFP, KMO, and CH25H for 24 h, respectively, followed by HSV-1 infection for 8 h

at MOI of 0.25, the titer of HSV-1 in the supernatant was measured by plaque assay. (D) 293T cells were transfected with siRNA for 24 h, followed by HSV-1 infection for 8 h at MOI of 0.25, the expression of KMO and HSV-1 UL27 was measured by RT-qPCR. (E) 293T cells were infected with shRNA-expressing lentivirus for 24 h, followed by HSV-1 infection for 8 h at MOI of 0.25, the expression of KMO and HSV-1 UL27 was measured by RT-qPCR. (F) 293T $kmo^{-/-}$ cell line was generated using the CRISPR/Cas9 system and confirmed DNA sequencing and Western Blotting Analysis. (G) 293T cells or 293T $kmo^{-/-}$ cells were infected with HSV-1 for 8 h at MOI of 0.25. Then, the expression of KMO was measured by RT-qPCR, and the expression of HSV-1-Luc was measured by Luciferase analysis. (H) $kmo^{-/-}$293T cells were transfected with plasmids (2 μg/mL) expressing GFP, KMO, CH25H for 24 h, followed by HSV-1 infection for 8 h at MOI of 0.25. Then, the expression of HSV-1 UL27 was measured by RT-qPCR. (I) The residues Tyr 99 (M1), Tyr 194 (M2), and Glu 366 (M3) were critical to the enzymatic activity of KMO, and we generated the activity-dead KMO mutants (KMO-M1-3) by site-directed mutagenesis. (J) 293T cells were transfected with plasmids (1 μg/mL) expressing GFP, KMO, KMO-M1, KMO-M2, KMO-M3 for 24 h respectively, followed by HSV-1 infection for 8 h at MOI of 0.25. Then, the HSV-1-luciferase and UL27 gene expression were measured by Luciferase analysis and RT-qPCR, respectively. (K) Vero cells and 293T cells were pretreated with conditional medium from the indicated construct-transfected 293T cells for 24 h, and then the pretreated cells were infected with HSV-1 at MOI of 0.25 for 24 h. Then, the expression of HSV-1 UL27 was measured by RT-qPCR. (L) 293T cells were pretreated with conditional medium from the indicated construct-transfected 293T cells for 24 h, and the pretreated cells were infected with HSV-1 at MOI of 0.25 for 24 h. Then, the expression of HSV-1 ICP27 protein was measured by Western Blotting Analysis. (M) RAW264.7 cells were pretreated with the indicated concentration of QUIN for 8 h and then infected with HSV-1 at MOI of 0.25 for 8 h. The expression of HSV-1-luciferase was measured by Luciferase analysis. The final data are presented as the mean ± SD of at least triplicate experiments. MOI: multiplicity of infection. $^*P < 0.05$, $^{**}P < 0.01$, $^{***}P < 0.001$.

cytotoxicity of QUIN, and the QUIN under 1 mM showed no obvious toxicity to different cell lines, including 293T, Vero, and Raw264.7 cells (S2 Fig).

## The antiviral mechanisms of KMO and QUIN by activating IRF3-mediated IFN-β production in HSV-1 infected cells

We next investigated the underlying mechanisms for the antiviral function of KMO and QUIN in HSV-1 infected cells. To clarify the possible roles of QUIN in antiviral immunity, we performed RNA-seq analyses for the HSV-1-infected RAW264.7 cells with or without QUIN treatment. Compared to the control group, QUIN treatment significantly altered the expression of 245 genes (161 upregulated and 84 downregulated) (S3A Fig). Gene ontology (GO) analyses showed that QUIN treatment upregulated the expression of multiple genes related to IFN-I production, cytokine production, cellular metabolic process (S3B Fig). KEGG analyses revealed enrichment in the Jak-STAT signaling pathway, Toll-like receptor signaling pathway, TNF signaling pathway (S3C Fig). After hierarchical clustering, we identified 19 genes involved in the antiviral signaling pathway by QUIN treatment (S3D Fig). These data suggested that QUIN and KMO might play the antiviral function by regulating IFN-I production and related signaling pathways and so on.

Consistent with these data, the over-expression of KMO significantly elicited the level of IFN-β expression in Raw 264.7 cells (Fig 3A). In contrast, siRNA or shRNA-mediated KMO knockdown and CRISPR/Cas9-mediated KMO knockout had significantly reduced the levels of IFN-β expression (Fig 3B–3D). Moreover, R061-8048, an inhibitor of KMO enzyme activity, could also inhibit the IFN-β production in a dose-dependent manner (Fig 3E). In addition, QUIN treatment significantly induced the IFN-β production in a dose-dependent manner (Fig 3F). These data suggested that induction of interferon production may contribute to the antiviral activity of KMO and QUIN.

We then studied the signaling pathway involved in KMO/QUIN-induced IFN-β production in HSV-infected cells and found that KMO over-expression effectively activated the phosphorylation of IRF3 at Ser396 (pSer396) (Fig 3G). Meanwhile, QUIN had a similar effect on the activation of IRF3 phosphorylation in a dose-dependent manner (Fig 3H). In contrast, KMO knockdown significantly decreased the phosphorylation of IRF3 in HSV-infected cells and consequently increased the expression of HSV-1 ICP27 protein (Fig 3I). Moreover, KMO inhibitor R061-8048 also suppressed the phosphorylation of IRF3 in response to HSV-1 infection (Fig 3J). Overall, these data indicated that KMO/QUIN promoted IFN-β production by enhancing IRF3 phosphorylation in HSV-infected cells.

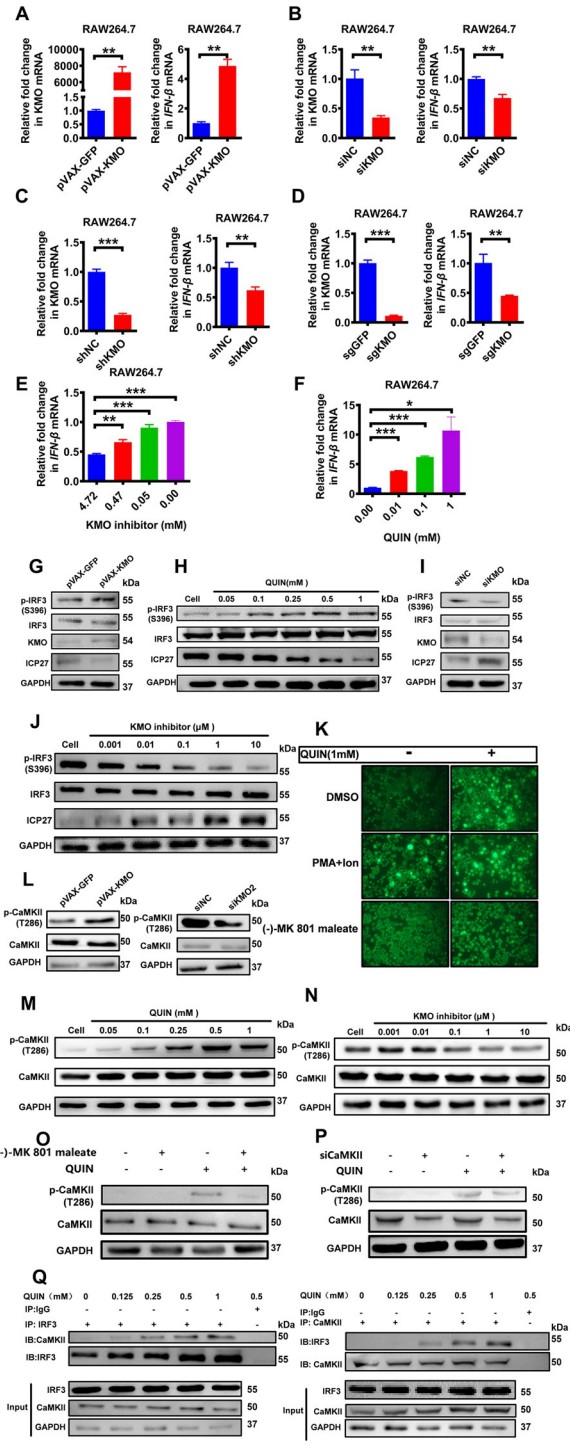

**Fig 3. The antiviral mechanisms of KMO and QUIN by activating IRF3 phosphorylation-mediated IFN-β production.** (A) RAW264.7 cells were transfected with plasmids (1 μg/mL) expressing GFP, KMO for 24 h, followed by HSV-1 infection at MOI of 0.25 for 8 h, and then the expression of KMO and IFN-β were measured by RT-qPCR. (B) RAW264.7 cells were transfected with siRNA for 24 h, followed by HSV-1 infection at MOI of 0.25 for 8 h, and the expression of KMO and IFN-β were measured by RT-qPCR. (C) RAW264.7 cells were infected with shRNA-expressing lentivirus for 24 h, followed by HSV-1 infection at MOI of 0.25 for 8 h, and the expression of KMO and IFN-β were measured by RT-qPCR. (D) RAW264.7 cells were infected with sgRNA-expressing lentivirus for 24 h, followed by HSV-1 infection at MOI of 0.25 for 8 h, and the expression of KMO and IFN-β were measured by RT-qPCR. (E) RAW264.7 cells were pretreated with the indicated concentration of KMO inhibitor for 8 h, followed by

HSV-1 infection at MOI of 0.25 for 8 h, and the expression of IFN-β was measured by RT-qPCR. (F) RAW264.7 cells were pretreated with the indicated concentration of QUIN for 8 h, followed by HSV-1 infection at MOI of 0.25 for 8 h, and the expression of IFN-β was measured by RT-qPCR. (G) Raw 264.7 cells were transfected with plasmids expressing GFP or KMO (1 μg/mL) for 24 h respectively, followed by HSV-1 infection at MOI of 0.25 for 8 h. Then, the expression levels of indicated proteins were detected by Western Blotting Analysis. (H) Raw 264.7 cells were pretreated with the indicated concentration of QUIN or DMSO for 8 h and then infected with HSV-1 at MOI of 0.25 for another 8 h. Then, the expression level of indicated proteins was detected by Western Blotting Analysis. (I) Raw 264.7 cells were transfected with siNC or siKMO for 24 h respectively, followed by HSV-1 infection at MOI of 0.25 for 8 h. Then, the expression levels of indicated proteins were detected by Western Blotting Analysis. (J) Raw 264.7 cells were pretreated with the indicated concentration of KMO inhibitor (R061-8048) or DMSO for 8 h, and then infected with HSV-1 at MOI of 0.25 for 8 h, the expression levels of indicated proteins were detected by Western Blotting Analysis. (K) Raw 264.7 cells were pretreated with NMDAR inhibitor ((-)-MK 801 maleate (10 μM)) or DMSO for 1h, and then stained with Calbryte 520 AM and stimulated with QUIN (1 mM), PMA (8 ng/mL) and ionomycin (200 ng/mL), or DMASO respectively. Then, the samples were imaged by Leica microscopy with the original magnification of ×40. (L) Raw 264.7 cells were transfected with plasmids (1 μg/mL) expressing GFP or KMO for 24 h respectively, followed by HSV-1 infection at MOI of 0.25 for 8 h. Then, the expression levels of indicated proteins were detected by Western Blotting Analysis. Moreover, Raw 264.7 cells were transfected with siNC or siKMO for 24 h, followed by HSV-1 infection at MOI of 0.25 for 8 h. Then, the expression levels of indicated proteins were detected by Western Blotting Analysis. (M) Raw 264.7 cells were pretreated with the indicated concentration of QUIN or DMSO for 8 h and then infected with HSV-1 at MOI of 0.25 for 8 h. Then, the expression level of indicated proteins was detected by Western Blotting Analysis. (N) Raw 264.7 cells were pretreated with the indicated concentration of KMO inhibitor (R061-8048) or DMSO for 8 h and then infected with HSV-1 at MOI 0.25 for 8 h. Then, the expression level of indicated proteins was detected by Western Blotting Analysis. (O) Raw 264.7 cells were pretreated with NMDAR inhibitor ((-)-MK 801 maleate (10 μM)) or DMSO for 1 h, and then treated with QUIN(1mM) or DMSO for 8 h. Then, the expression level of indicated proteins was detected by Western Blotting Analysis. (P) Raw 264.7 cells were transfected with siNC or siCaMKII for 24 h, followed by QUIN(1mM) or DMSO treatment for 8 h, and then the expression level of indicated proteins were detected by Western Blotting Analysis. (Q) Raw 264.7 cells were pretreated with the indicated concentration of QUIN or DMSO for 8 h. Cell extracts were immunoprecipitated (IP) with anti-IRF3 or anti-CaMKII antibody. IRF3-conjugated proteins were detected with anti-CaMKII antibody, and CaMKII-conjugated proteins were detected with anti-IRF3 antibody by Western Blotting (top). The whole-cell extracts were also detected by Western Blotting using the indicated antibody (bottom). The final data are presented as the mean ± SD of at least triplicate experiments. $^*P < 0.05$, $^{**}P < 0.01$, $^{***}P < 0.001$.

Subsequently, we explored how KMO/QUIN modulated the IRF3 phosphorylation. QUIN was previously reported as an agonist of NMDAR to induce the $Ca^{2+}$ influx [34,35]. In our study, we first validated the NMDAR expression in various cell lines (S4 Fig), and then we determined whether QUIN could elicit $Ca^{2+}$ influx. Using Calbryte 520 to visually label the intracellular $Ca^{2+}$, an immediate increase of the intracellular $Ca^{2+}$ was observed after QUIN stimulation in Raw 264.7 cells, while this increase of $Ca^{2+}$ influx was significantly suppressed in the samples treated with (-)-MK-801 (Fig 3K), which is an NMDAR inhibitor [36].

Considering that the increased $Ca^{2+}$ influx can trigger IFN-I production through the Calcium/calmodulin-dependent protein kinase II (CaMKII) and IRF3 signaling pathway [37], we further determined whether KMO/QUIN modulated the phosphorylation of CaMKII, which is a major biochemical decoder of intracellular $Ca^{2+}$ oscillations [38]. Results showed that KMO over-expression enhanced while KMO knockdown reduced the CaMKII phosphorylation (T286) in HSV-1 infected cells (Fig 3L). A similar observation of increased CaMKII phosphorylation (T286) was also observed with QUIN incubation (Fig 3M). In contrast, the QUIN-induced CaMKII phosphorylation was significantly inhibited in the presence of the KMO inhibitor (Ro 61–8048) (Fig 3N), (-)-MK-801(Fig 3O) or siCaMKII (Fig 3P). We have demonstrated a direct interaction between CaMKII and IRF3 in QUIN-stimulated cells in a dose-dependent manner (Fig 3Q). Taken together, KMO/QUIN induced the production of IFN-I via activating the NMDAR/$Ca^{2+}$ influx to trigger the CaMKII/IRF3 signaling pathway.

## Broadly antiviral activity of KMO and QUIN

As KMO had a novel function against viral infection by enhancing IFN-I induction, we wanted to determine the breadth of antiviral activity of KMO. Our results demonstrated that KMO

could effectively inhibit the replication of DNA viruses, such as HSV-1 (a kind of enveloped DNA virus) and adenovirus (a kind of nonenveloped DNA virus) in a dose-dependent manner (Fig 4A and 4B). Moreover, we also found that KMO had a significant antiviral effect against RNA viruses, such as negative-strand RNA viruses, including vesicular stomatitis virus (VSV) and influenza virus (PR8) (Fig 4C and 4G), and positive-stranded RNA viruses, including Zika virus (ZIKV), dengue virus (DENV) and SARS-CoV-2 (Fig 4D and 4F). In addition, we found that KMO significantly enhanced IFN-β expression during infection with the viruses mentioned above (S5A Fig). Similarly, QUIN treatment also had dose-dependent inhibitory effects on these viruses by stimulating the production of IFN-β (Figs 4H–4N, and S5B). We found that QUIN also has broad antiviral activity in various cells, including A549 and BMM cells (S6 Fig). Thus, our data showed that KMO/QUIN exerted an unreported function to inhibit DNA viruses and RNA viruses broadly.

Previous studies have reported that IDO1 played a function in inhibiting viral replication [39,40]. To clarify that KMO and QUIN have antiviral activities independent of IDO1, we treated cells with 1-Methyl-D-tryptophan(1-MT), an inhibitor of IDO1. We found that 1-MT treatment weakened the antiviral effect by inhibiting IDO1 (S7A and S7B Fig). However, KMO and QUIN treatment could rescue the antiviral effect even after the IDO1 inhibition by 1-MT (S7C and S7D Fig), indicating that the antiviral effects of KMO and QUIN were independent on IDO1.

## QUIN treatment protected mice from viral infections

Next, we sought to investigate the efficacy of QUIN as a potential antiviral agent against viral infections *in vivo*. Mice were infected with highly pathogenic HSV-1 McKrae strain, followed by daily treatment with QUIN for 7 days (Fig 5A). We also used the acyclovir (ACV) as a positive control, a clinical drug prescribed to control herpes infection. At 7 days post-infection (dpi.), the vehicle group's progressive corneal scarring and visual impairment deteriorated significantly, while this symptom was greatly alleviated in the QUIN and the ACV groups (Fig 5B). More importantly, QUIN administration effectively protected mice and significantly improved their survival when challenged with the highly pathogenic HSV-1 McKrae strain, while all of the mice in the vehicle group succumbed to this lethal challenge within 7 dpi (Fig 5C). Consistent with this finding, mice treated with QUIN had significantly lower disease scores when compared to the vehicle group (Fig 5D).

In addition, the viral copies in the eye washing fluid from QUIN-treated mice were significantly reduced than that of mock-treated mice (Fig 5E). We also quantitatively measured the concentration of IFN-β in serum, brain, and spleen of these experimental mice and found a significant increase of IFN-β secretion in the QUIN-treated mice than that of mock-treated mice (Fig 5F). Furthermore, there was an increased frequency of HSV-1 antigen-specific IFN-γ-secreting cells, determined by the enzyme-linked immunosorbent spot (ELISPOT) assays, in the QUIN group compared to that of the vehicle group (Fig 5G), implying that QUIN therapy also modulated the adaptive immune responses, especially the antigen-specific cellular immune responses. Collectively, these data indicated that QUIN treatment effectively protected mice from viral infection.

## *kmo⁻ᐟ⁻* mice are more susceptible to viral infections

Finally, to determine whether KMO played a physiological role in host defense against viral infections, we generated the *kmo⁻ᐟ⁻* mice by CRISPR/Cas9-based strategy (S8 Fig) and then compared whether there was an increased susceptibility to viral infections in *kmo⁻ᐟ⁻* mice as compared to the age-matched wild-type mice (*kmo⁺ᐟ⁺*). After challenging with sub-lethal

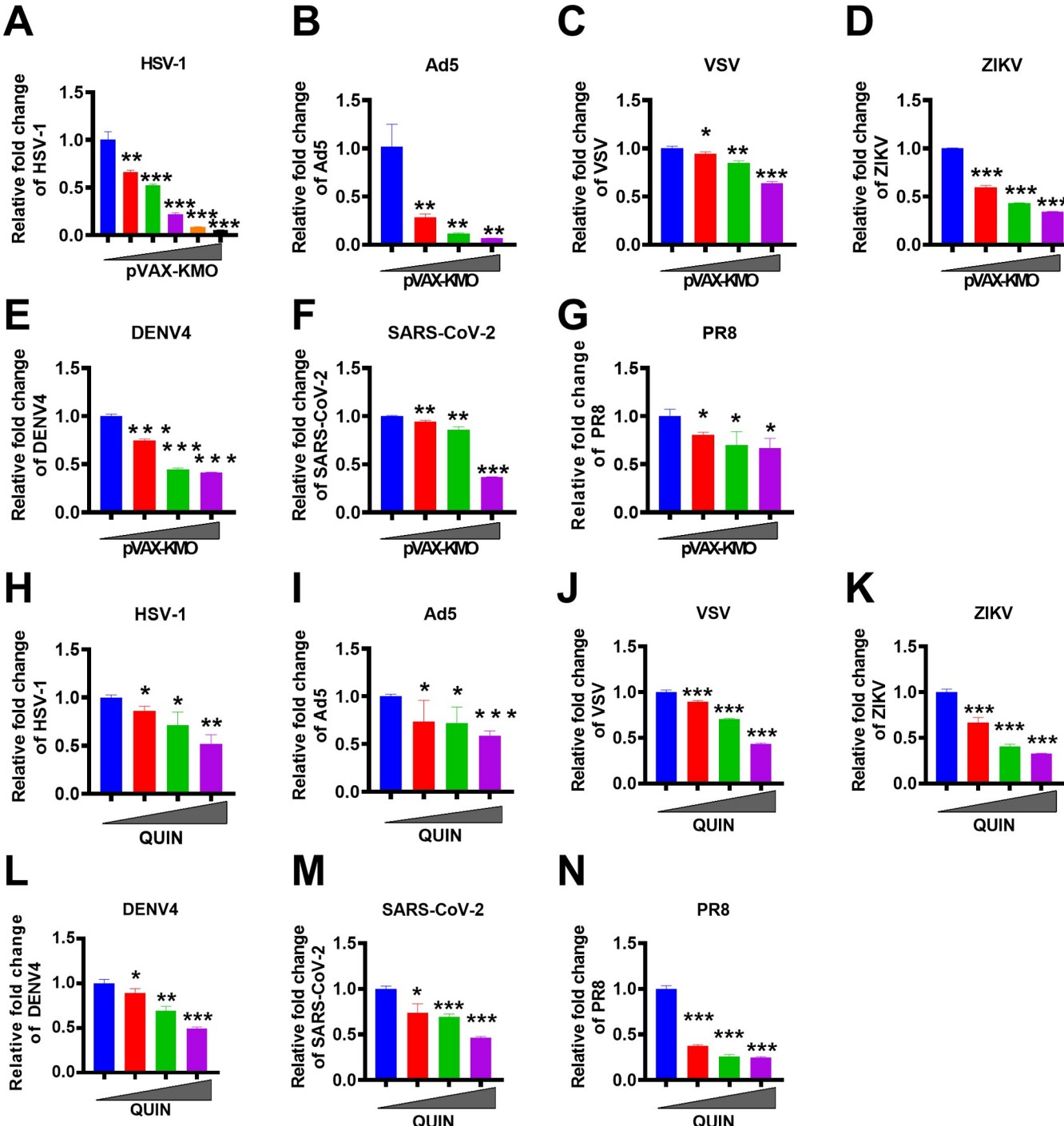

**Fig 4. Broadly antiviral activities of KMO and QUIN.** (A) 293T cells were transfected with different concentrations of KMO-expressing plasmid for 24 h, followed by HSV-1 infection at MOI of 1 for 8 h. The expression level of HSV-1 UL27 was quantified by RT-qPCR. (B) 293T cells were transfected with different concentrations of KMO-expressing plasmid for 24 h, followed by Ad5 infection at MOI of 1 for 8 h. The expression level of Ad5 was quantified by RT-qPCR. (C) 293T cells were transfected with different concentrations of KMO-expressing plasmid for 24 h, followed by VSV infection at MOI of 1 for 8 h. The expression level of VSV was quantified by RT-qPCR. (D) 293T cells were transfected with different concentrations of KMO-expressing plasmid for 24 h, followed by ZIKV infection at MOI of 1 for 8 h. The expression level of ZIKV was quantified by RT-qPCR. (E) 293T cells were transfected with different concentrations of KMO-expressing plasmid for 24 h, followed by DENV4 infection at MOI of 1 for 8 h. The expression level of DENV4 was quantified by RT-qPCR. (F) CoCa-N-2 cells were transfected with different concentrations of KMO-expressing plasmid for 24 h, followed by replication-competent SARS-CoV-2 virus-like-particles (SARS-CoV-2 GFP/ΔN) infection at MOI of 1 for 8 h. The expression level of SARS-CoV-2 GFP/ΔN was quantified by RT-qPCR. (G) 293T cells were transfected with different concentrations of KMO-expressing plasmid for 24 h, and the expression level of PR8 was quantified by RT- qPCR.

(H-N) Cells were pretreated with QUIN at different concentrations for 8 h, followed by viral infections for 8 h at MOI of 1, including HSV-1, VSV, ZIKV, DENV4, Ad5, SARS-CoV-2 GFP/ΔN (SARS-CoV-2) and PR8, and then the expression level of corresponding viruses was quantified by RT-qPCR. The expression level of mRNA was normalized to the expression of β-actin, and the data from at least triplicates were shown as the mean ± SD. $^*P < 0.05$, $^{**}P < 0.01$, $^{***}P < 0.001$.

doses of highly pathogenic HSV-1 McKrae strain (Fig 6A), $kmo^{-/-}$ mice developed a more severe disease progression and clinical symptoms, leading to a significantly lower survival rate than that of $kmo^{+/+}$ mice (Fig 6B). Furthermore, plaque assay indicated that the viral copies in the eye washing fluid from $kmo^{-/-}$ mice were significantly higher than $kmo^{+/+}$ mice (Fig 6C). Our data also showed that peritoneal and bone marrow-derived macrophages from the $kmo^{-/-}$ mice were more susceptible to HSV-1 infection than that of $kmo^{+/+}$ mice (Fig 6D). Consistent with the above *in vitro* data (Fig 3D), the level of IFN-β expression in serum, brain, and spleen of $kmo^{-/-}$ mice was significantly lower than that of $kmo^{+/+}$ mice (Fig 6E and 6F). In addition, we further showed that there was an obvious dysfunction of adaptive T cell immune responses in $kmo^{-/-}$ mice, characteristic with the decreased frequency of the polyfunctional CD4+ T cells and CD8+ T cells secreting interferon-gamma (IFN-γ), tumor necrosis factor-alpha (TNF-α), and interleukin-2 (IL-2) cytokines (Fig 6G).

We also performed immunohistochemistry and hematoxylin-eosin (H&E) staining to examine the pathological changes in the tissues of these experimental mice. Compared with $kmo^{+/+}$ mice, the McKrae HSV-1 infected $kmo^{-/-}$ mice had the increased cytoplasmic eosinophilia (black arrow) and focal infiltration of inflammatory cells (red arrow) in the liver lobules, and the increased neutrophil infiltration in the red pulp of the spleen (black arrow) (Fig 6H). Importantly, an obvious pathological injury was observed in the trigeminal ganglia in $kmo^{-/-}$ mice, characteristic with the increased neuron necrosis (black arrow), decreased neuronal soma count, enhanced inflammatory cell infiltration (red arrow), and widened the gap between neuronal soma (yellow arrow) (Fig 6H). This status of immune imbalance in $kmo^{-/-}$ mice was also confirmed by complete blood count analysis, and the results indicated that the numbers of white blood cell (WBC), lymphocyte (Lym), monocyte (Mon), granulocyte (Gran) were significantly decreased in $kmo^{-/-}$ mice than those of $kmo^{+/+}$ mice (Fig 6I).

We next performed RNA-seq analyses for the HSV-1-infected BMM cells from $kmo^{+/+}$ and $kmo^{-/-}$ mice. Compared to the $kmo^{-/-}$ group, the $kmo^{+/+}$ group significantly altered the expression of 2929 genes (975 upregulated and 1954 downregulated) (S9A Fig). Gene ontology (GO) analyses showed that the $kmo^{+/+}$ group upregulated the expression of multiple genes related to IFN-β and IFN-γ production, immune system process, defense response (S9B Fig). KEGG analyses revealed enrichment in the Jak-STAT signaling pathway, Toll-like receptor signaling pathway, TNF signaling pathway (S9C Fig). After hierarchical clustering, we identified 25 genes involved in the antiviral signaling pathway (S9D Fig). These data suggested that $kmo$ knockout might reduce the antiviral function by regulating IFN-I production and related signaling pathways.

To further clarify whether QUIN has a rescue effect on virus-infected $kmo^{-/-}$ mice, $kmo^{+/+}$ and $kmo^{-/-}$ mice were challenged with HSV-1 McKrae strain, followed by daily treatment with different concentrations of QUIN for 7 days (S10A Fig). QUIN administration effectively protected the mice and reduced disease scores in a dose-dependent manner (S10B and S10C Fig). The viral copies in the eye washing fluid from QUIN-treated (2.5 mg/kg) mice were significantly reduced than that of mock-treated mice. Of note, in the $kmo^{+/+}$ group, the viral copies were significantly reduced in the QUIN-treated group at a lower dose of 1 mg/kg (S10D Fig). In addition, IFN-β secretion was significantly increased in the serum of QUIN-treated mice (S10E Fig).

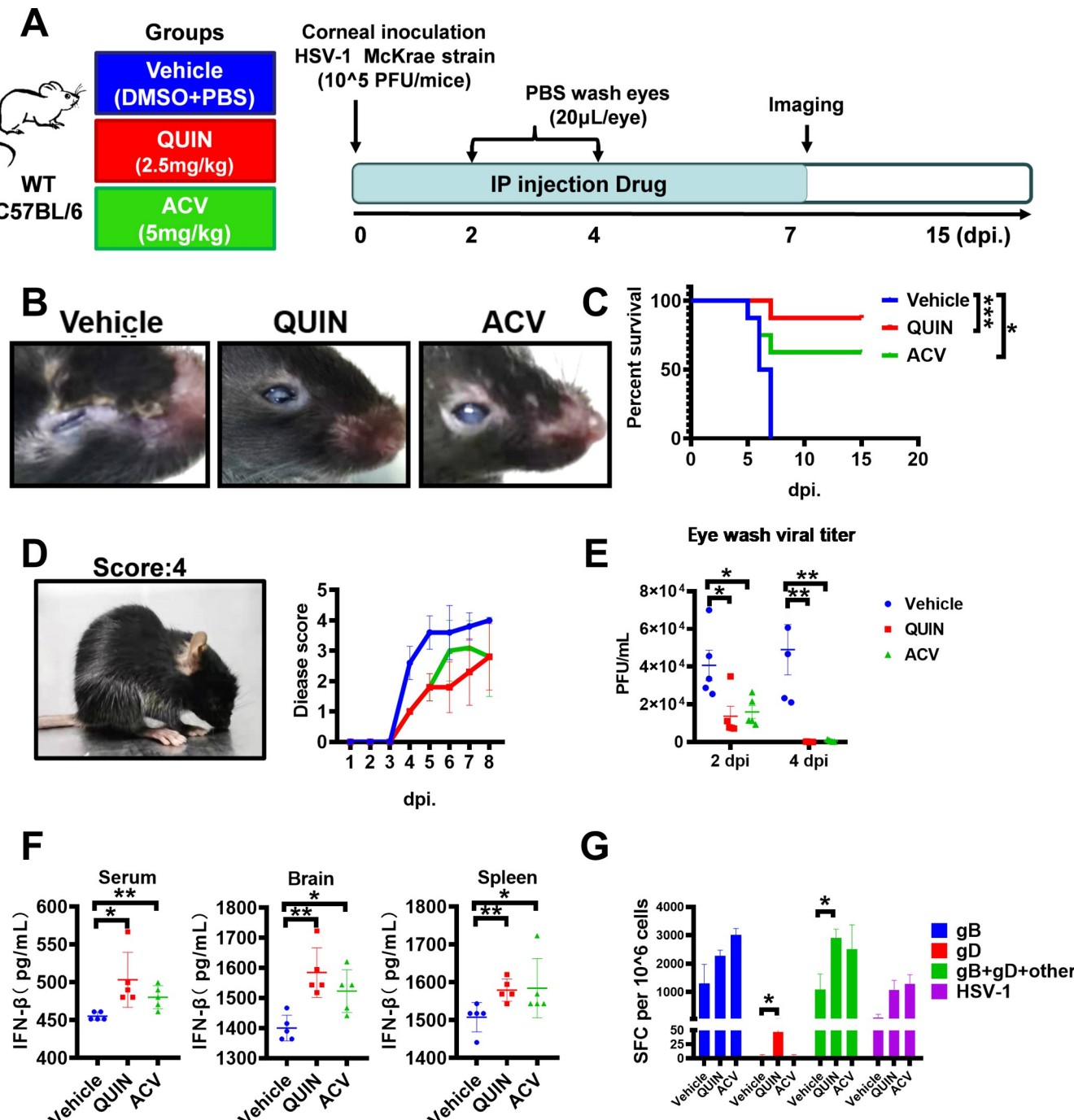

**Fig 5. QUIN treatment protected mice from viral infections.** (A) Schedule for evaluating the therapeutic efficacy of QUIN in mice. Briefly, C57BL/6 mice were corneally inoculated with $10^5$ PFU HSV-1 McKrae strain and treated daily with QUIN for 7 days. The vehicle group (negative control) was received intraperitoneal (IP) injection of DMSO and PBS. QUIN group was received an IP injection of QUIN (2.5 mg/kg). The ACV group (positive control) received an IP injection of ACV (5 mg/kg). The eyes of mice were washed at 2 dpi and 4 dpi to perform the plaque assay. The disease symptoms of experimental mice were monitored until 15 dpi. (B) The representative picture of progressive corneal scarring, visual impairment of experimental mice in different groups at 7 dpi. (C) Survival curve of experimental mice in different groups over time post-infection (n = 8). (D) Statistical analysis of disease scores of experimental mice in different groups over time (n = 5). Score 0, healthy; Score 4, being severe disease. (E) The HSV-1 titer in the eye washing fluid at 2 dpi and 4 dpi were measured by plaque assay (n = 5 per group). (F) The concentration of IFN-β in serum, brain, and spleen of experimental mice at 5 dpi was measured by ELISA assay (n = 5 per group). (G) The frequency of HSV-1 antigen-specific IFN-γ-secreting cells was determined by the enzyme-linked immunosorbent spot (ELISPOT) assays at 5 dpi. Data represents spot-forming cells (SFC) per million cells. The final data are presented as the mean ± SD of triplicate experiments. $^*P < 0.05$, $^{**}P < 0.01$, $^{***}P < 0.001$.

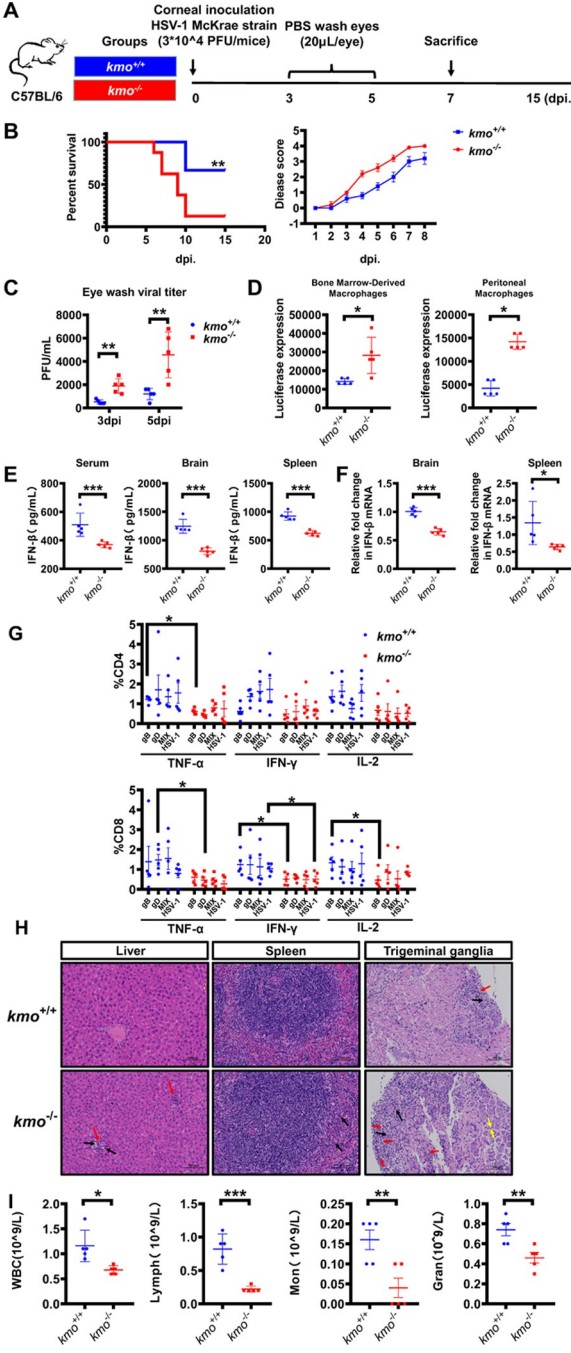

**Fig 6. *kmo*⁻ᐟ⁻ mice are more susceptible to viral infections.** (A) Schedule for evaluating the susceptibility to viral infections in $kmo^{-/-}$ mice. Briefly, both $kmo^{+/+}$ C57BL/6 mice and $kmo^{-/-}$ C57BL/6 mice were corneally inoculated with $3\times10^4$ PFU HSV-1 McKrae strain, disease symptoms of experimental mice were monitored until 15 dpi. (B) Survival curve and disease score of experimental mice in different groups over time post-infection (n = 8 per group). (C) The HSV-1 titer in the eye washing fluid at 3 dpi and 5 dpi was measured by plaque assay (n = 5 per group). (D) Bone Marrow-derived and Peritoneal macrophages were isolated from $kmo^{+/+}$ or $kmo^{-/-}$ mice and then infected with HSV-1-Luc at MOI of 0.25 for 8 h. Then, the luciferase expression was measured by Luciferase Assay kit (n = 5 per group). (E) The concentration of IFN-β in serum, brain, and spleen of experimental mice at 7 dpi was measured by ELISA assay (n = 5 per group). (F) The expression of IFN-β in the brain and spleen of $kmo^{+/+}$ or $kmo^{-/-}$ mice were measured by RT-qPCR (n = 5 per group). (G) The frequency of polyfunctional CD4+ and CD8+ T cell populations in $kmo^{+/+}$ or $kmo^{-/-}$ mice was assessed by detecting the secretion of TNF-α, IFN-γ, and IL-2 cytokines response to HSV-1 peptide pool stimulation. (H) The pathological changes of experimental mice's liver, spleen, and trigeminal ganglia (TG) were observed by hematoxylin-eosin (H&E) staining at 7dpi. The scale bars: 100 μm. (I) The numbers of white blood cell

(WBC), lymphocyte (Lym), monocyte (Mon), granulocyte (Gran) in HSV-1 infected *kmo*$^{-/-}$ mice and *kmo*$^{+/+}$ mice were detected by complete blood count analysis at 7 dpi (n = 5 per group). The final data are presented as the mean ± SD of at least triplicate experiments. $^*P < 0.05$, $^{**}P < 0.01$, $^{***}P < 0.001$.

Overall, these findings indicated that KMO is a key antiviral factor physiologically involved in modulating antiviral immunity.

## Discussion

Independent of its known regulatory role on tryptophan metabolism, we herein reported that KMO and its enzymatic product QUIN exerted a novel broad-spectrum antiviral function against numerous emerging pathogenic viruses, thorough triggering the NMDAR/Ca$^{2+}$ influx and CaMKII/ IRF3-mediated IFN-β production. Previously, KMO had been shown to associate with pathological conditions like tumorigenesis [41–43] and neurodegenerative diseases [44–48]. However, our study reported its new function in host immunity against viral infections. Importantly, in the animal infection model, QUIN treatment effectively inhibited viral infections and alleviated disease progression *in vivo*. Moreover, the *kmo* gene is a physiologically key antiviral factor because *kmo*$^{-/-}$ mice were more vulnerable to pathogenic viral challenge than *kmo*$^{+/+}$ mice (Fig 7). As a result, our findings revealed the critical role of tryptophan metabolism in antiviral immunity, which highlighted the importance of further understanding the profound intricacy between innate immunity and cellular metabolism.

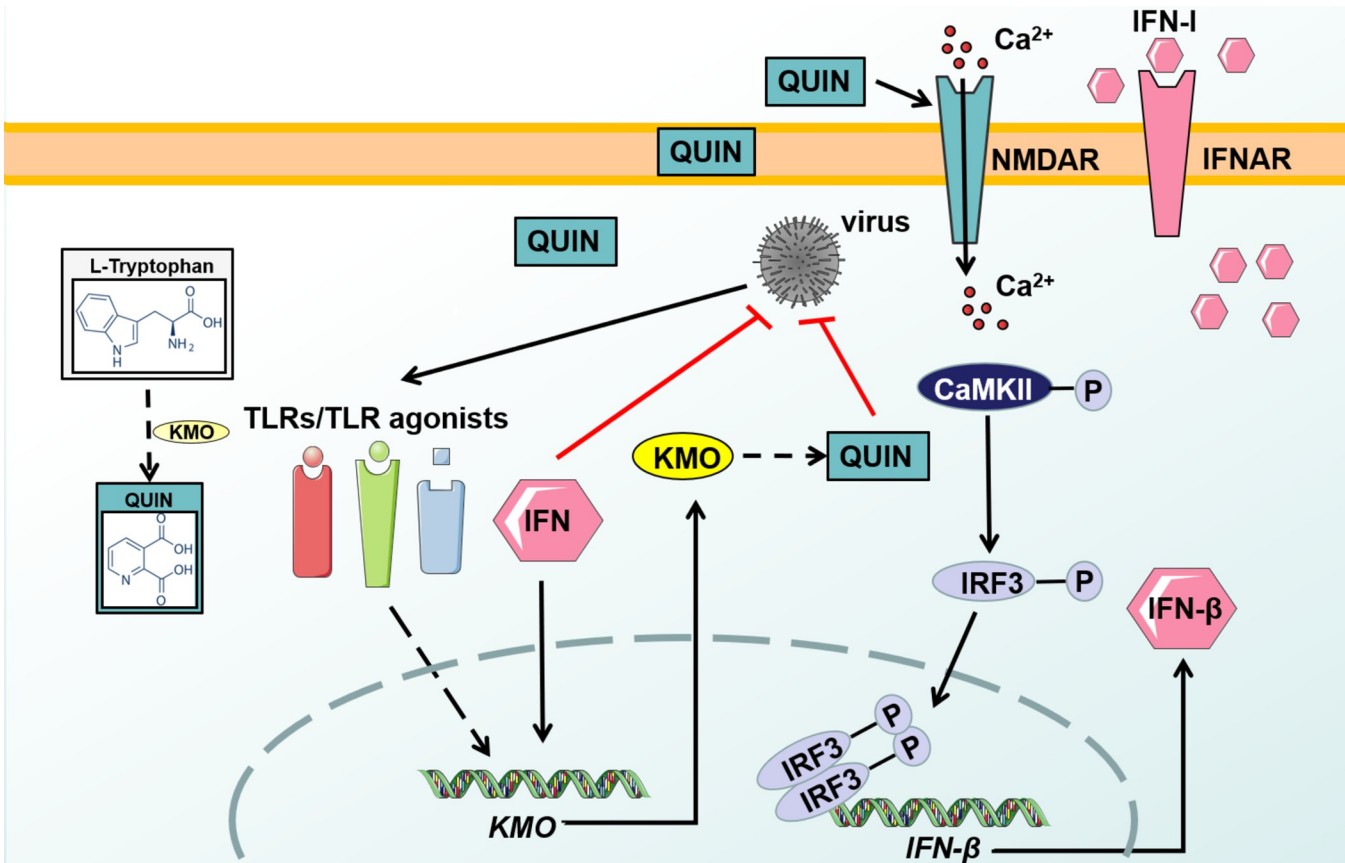

**Fig 7. A schematic illustrating KMO and its enzymatic product QUIN exerted a novel broadly antiviral function.** QUIN triggered the NMDAR/Ca$^{2+}$ influx and CaMKII/IRF3-mediated IFN-β production.

The tryptophan metabolism, mainly processed by the kynurenine pathway (KP), is an essential physiological process to maintain our body's health, and the key rate-limiting enzymes in the KP include IDO1, IDO2, TDO, and KMO [49]. Recently, KP has also been reported to play a critical role in modulating immune homeostasis [15], and the KP metabolites might be associated with immune disorders [50–52]. For example, IDO1 could inhibit the replication of human cytomegalovirus, HSV-1, and retroviruses through the increased consumption of L-Try [53–55]. However, the role of KMO in modulating innate immunity against viral infections has not been investigated. Our study examined the serum of $kmo^{+/+}$ and $kmo^{-/-}$ mice. We found that kynurenine accumulated because of the absence of kmo (S11 Fig), which might have a negative feedback inhibition on IDO1. Still, our further research showed that the KMO and QUIN could independently inhibit the virus when IDO1 was inhibited. In addition, IDO-mediated tryptophan consumption was not involved in the IFN-α/β-mediated antiviral effects [40], but the antiviral effects of KMO and QUIN in our study were associated with IFN-β production. Therefore, the antiviral effects of KMO and IDO1 may have a synergistic relationship through different antiviral pathways.

Our study found that the clinical symptoms in pathogenic HSV-1 infected $kmo^{-/-}$ mice were more severe than those in $kmo^{+/+}$ mice, and this observation was consistent with a lower IFN-γ secretion in $kmo^{-/-}$ mice than in $kmo^{+/+}$ mice. One study showed that KMO deficiency increased the frequency of Foxp3+ regulatory T cells and the levels of anti-inflammatory cytokines, including transforming growth factor-β and interleukin-10 [56]. On the contrary, another study showed that KMO deficiency reduced the mortality in encephalomyocarditis virus-infected mice [57]. Therefore, KMO might play dual roles in regulating immune responses through various mechanisms, thus exerting beneficial and harmful effects on various diseases. Further studies are consequently needed to clarify how KMO precisely modulates inflammation, innate immunity, and adaptive immunity, especially in patients with autoimmune diseases and chronically infectious diseases.

Tryptophan metabolism through the KP is often preferentially directed to produce QUIN catalyzed by KMO [50]. Our studies indicated that the antiviral activity of KMO was mediated through its enzymatic product QUIN. We have provided evidence that QUIN activates the N-methyl-d-aspartate receptor (NMDAR) and $Ca^{2+}$ influx, which trigger the phosphorylation of CaMKII and IRF3, leading to IFN-I production. The previous study also found that $Ca^{2+}$ and its major downstream effector, CaMKII, are important for immune cells' functions, and CaMKII could significantly enhance the production of IFN-α/β in macrophages [37,58]. However, other studies demonstrated that NMDAR antagonists such as MK-801 inhibited Japanese encephalitis virus (JEV)-mediated neuropathogenesis and VSV infection [59,60]. Therefore, modulation of NMDAR in different cells or tissues might be a double-edged sword.

Of note, besides the enhanced IFN-β-mediated innate immunity, our data showed that QUIN treatment also promoted the IFN-γ-mediated cytolytic T lymphocytes (CTL) against HSV-1 antigen-specific stimulation. Stimulating antigen-specific CTL is essential for eradicating virus-infected host cells [61] and inhibiting the activation of viral latency [62], which is important for controlling chronically viral infections, such as HIV-1, HSV-1. Especially, IFN-γ could effectively suppress the reactivation from HSV-1 latency in sensory neurons [63,64]. Therefore, the enhancement of HSV-1-specific CTL responses by QUIN treatment contributed to suppressing herpes disease progression and providing long-term protection in our study.

The structural skeleton of QUIN and its derivatives were previously reported as an important pharmacophore and explored to develop anti-infectious agents, including anti-malaria [65,66], anti-tuberculosis [67], and anti-tumor [68]. Our studies showed that QUIN had broad antiviral activities against a broad range of DNA and RNA viruses, including HSV-1,

adenovirus, VSV, influenza virus (PR8), ZIKV, DENV, and SARS-CoV-2. We further demonstrated that administration of QUIN significantly inhibited viral infections and alleviated disease progression in the mouse model of pathogenic HSV-1 infection. To be noted, QUIN is a neuroactive metabolite and can induce oxidative neurotoxicity [69–71]. A high level of QUIN was reported to associate with Alzheimer's disease, anxiety, depression, epilepsy, and Huntington's disease [72]. Therefore, its potential neurotoxicity should be addressed when developing the QUIN as an antiviral drug candidate. Since QUIN cannot penetrate the blood-brain barrier [73], and thus QUIN was injected intraperitoneally to exert antiviral effects in our *in vivo* experiment to avoid the potential side effect. Nevertheless, the safety of the long-term administration of QUIN should be further monitored in the future.

Some studies reported that the concentration of the QUIN was increased because of KP activation in response to some pathogen infections (such as HIV-1, HSV-1, influenza A virus) [74–80]. Importantly, recent studies also reported that KP metabolites, including kynurenate, kynurenine, 8-methoxykynurenate, were enriched in COVID-19 patients, and the decrease in tryptophan and the increase in kynurenine were correlated to the COVID-19 disease severity [81–83]. Given that we have encountered numerous challenges with emerging pathogenic viruses in the past decades years, including SARS, MARS, EBOLA, ZIKV, and SARS-COV-2, further studies on immune regulation of metabolic genes such as KMO may provide insights to develop novel broad antiviral agents like QUIN to treat diseases associated with many unpredictable, viral pathogens.

In summary, it is of great interest to further investigate the possibility of applying KMO and QUIN as antiviral drug candidates in future studies.

## Materials and methods

### Ethics statement

The animal study was approved by the Institutional Review Boards and Animal Care and Use Committees of Sun Yat-Sen University (Approval No. SYSU-IACUC-2021-000048).

### Cell lines

293T cells (from the embryonic kidney of a female human fetus), Vero cells (from the kidney of a female normal adult African green monkey), A549 cells (from human alveolar adenocarcinoma basal epithelial cells), Hela cells (from human cervical cancer tissue)and RAW 264.7 cells (from macrophage of a male adult mouse) were cultured in complete Dulbecco's modified Eagle's medium (DMEM, Gibco) containing 10% fetal bovine serum (FBS, Gibco) and 1% penicillin/streptomycin (Gibco), at 37˚C in an atmosphere of 5% of $CO_2$. THP-1 cells (from the peripheral blood of a boy with acute monocytic leukemia) were cultured in conditioned RPMI 1640 medium containing 10% fetal bovine serum (FBS, Gibco), 1% penicillin/streptomycin (Gibco) at 37˚C in an atmosphere of 5% of $CO_2$. The above cells were stocked in our laboratory.

293T $kmo^{-/-}$ cells were constructed in our lab. In brief, 293 T cells were infected with sgRNA-expressing lentivirus with polybrene and then added with puromycin. The single clone of the $kmo^{-/-}$ cell line was selected and validated by Western Blotting analysis and DNA sequencing.

Wild-type bone-marrow-derived macrophage cells (WT-J2-BMM) and interferon-α receptor-deficient cells ($Ifnar^{-/-}$J2-BMM) were gifted by Genhong Cheng (UCLA, USA) and cultured in conditioned RPMI 1640 medium containing 10% fetal bovine serum (FBS, Gibco), 1% penicillin/streptomycin (Gibco), 10 mM HEPES (pH 7.8), and 1% M-CSF (Gibco).

The $kmo^{-/-}$ bone marrow-derived and peritoneal macrophages were isolated from $kmo^{-/-}$ mice as previously described [84]. Briefly, bone marrow-derived macrophages were isolated from femurs and tibias of $kmo^{-/-}$ mice. Peritoneal macrophages isolation is performed by

injecting 4 mL PBS 1× through the peritoneal wall into the peritoneal, gently massaging the mouse abdomen, slowly recovering as much PBS 1× as possible. Cells were cultured in 10 mL conditioned medium (DMEM medium (Corning) with 10% fetal bovine serum, 1% L gluta-mine, 1% penicillin-streptomycin (Gibco), 100 ng/mL M-CSF (Biolegend) at 37˚C in an atmo-sphere containing 5% $CO_2$ in an incubator. On day 3, most cells were adherent to the dish. Discard the medium from the dish, and then the dish was supplemented with 10 mL of condi-tioned medium for further growth until day 6. On day 7, the cells were used to perform the corresponding experiments.

## Viruses

VSV-GFP (VSV) was gifted by Dr. Tian Lan (VectorBuilder, Guangzhou, China). DENV4 and ZIKV were kindly gifted by Dr. Zhongyu Liu (Sun Yat-sen University, Guangzhou, China). Influenza virus A/PR8 was gifted by Prof. Yuelong Shu (Sun Yat-sen University, Guangzhou, China). Replication-competent SARS-CoV-2 virus-like-particles (SARS-CoV-2 GFP/ΔN) and Caco-2 cells expressing N protein (Caco-2- N) were gifted by Prof. Qiang Ding (Center for Infectious Disease Research, School of Medicine, Tsinghua University, Beijing, China). Ad5-GFP and HSV-GFP-Luc were stored in our lab. These viruses were used in cell studies.

The pathogenic HSV-1 McKrae strain (McKrae) was gifted by Prof. Jumin Zhou (Kunming Institute of Zoology, Chinese Academy of Sciences. Kunming, China). McKrae was used in an animal study.

## Mice

WT C57BL/6 mice were purchased from the Laboratory Animal Resource Center of Sun Yat-sen University and bred in the SPF animal facility of Sun Yat-sen University in individually ventilated cages.

kmo-deficient C57BL/6 mice ($kmo^{-/-}$ mice) were constructed by Cyagen Biosciences and bred in the SPF animal facility of the Laboratory Animal Resource Center of Sun Yat-sen Uni-versity in individually ventilated cages. Briefly, using CRISPR/Cas9 technology, we obtained the *kmo*-knockout mice by applying high-throughput electro-transformation of sgRNA to fer-tilized eggs. The heterozygous F1 mice were obtained after being bred with wild-type C57BL/6 mice, and then the homozygous $kmo^{-/-}$ F2 mice after being bred with heterozygous F1 mice. The tail tip was cut, and the tissue DNA was extracted to perform a PCR assay to identify the homozygous $kmo^{-/-}$ mice (S8 Fig). In addition, the PCR products were also be sequenced for further confirmation.

## Plasmid constructs

pVAX-KMO, pVAX-CH25H, pVAX-GFP: The full-length human KMO and human CH25H were cloned into the pVAX vector. pVAX-Green fluorescent protein (GFP) plasmid was stored in our laboratory used as a mock transfection in our study. KMO hydroxylase activity-dead mutant (KMO-M) was generated by site-directed mutagenesis kit (TransGen Biotech) from pVAX-KMO construct as described above. The primers were listed in the S1 Table.

## Lentivirus production

**Small guide RNA (sgRNA).** sgRNAs targeting *kmo* gene were designed by using a CRISPR design tool (http://zlab.bio/guide-design-resources)), and three highest-scoring oligo-nucleotides were selected and cloned into pLentiCRISPRv2 (Addgene #52961) respectively. The recombinant lentivirus was generated by co-transfecting 293T cells with a cocktail of

lentiCRISPRv2, packaging plasmid pMD2.G (Addgene #12259) and psPAX2 (Addgene #12260) using Lipofectamine 2000 transfection reagent (Invitrogen). The primers were listed in the S1 Table.

**Short hairpin RNA (shRNA).** shRNAs targeting KMO mRNA were designed using an shRNA design tool (http://sirna.wi.mit.edu/), and the top-scoring targets were selected in this study. The forward and reverse oligonucleotides were synthesized and then annealed and cloned into the pLKO.1 vector. Lentiviral vectors were generated by co-transfecting 293T cells with a cocktail of pLKO.1 shRNA plasmid, packaging plasmid psPAX2, envelope plasmid pMD2.G using Lipofectamine 2000 transfection reagent (Invitrogen). The primers were listed in the S1 Table.

## siRNA and cell transfection

According to the manufacturer's manuals, the cultured cells with 80% confluence were transfected with different plasmids using Lipofectamine 2000 Transfection Reagent. The medium was replaced with DMEM containing 5% FBS and 1% penicillin/streptomycin after 5 h transfection, and then the cells were incubated for 24 h or 48 h.

The siRNA oligonucleotide duplexes targeting KMO were synthesized by Sangon (Shanghai, China). The negative-control siRNAs were purchased from Sangon. According to the manufacturer's protocol, the cells were transfected with 100 nM of the indicated siRNAs for 48–72 h by using Lipofectamine RNAiMax transfection reagent (Invitrogen). The knockdown efficacy of the target genes was detected by quantitative real-time PCR (RT-qPCR) or Western Blotting Analysis. The sequences of all siRNAs were listed in S1 Table, and all primers for RT-qPCR were listed in the S2 Table.

## Western Blotting analysis

The Western Blotting assay was performed as previously described [27]. Reagents and antibodies are listed in S4 Table.

## RNA-seq library preparation, sequencing, and data processing

RNA-seq experiments were performed as previously described [85]. In brief, RAW264.7 cells were incubated with or without QUIN at a dose of 1mM for 8 hours. Then the cells were infected with HSV-1 for another 8 hours. Bone marrow-derived macrophages from the $kmo^{+/+}$ mice and $kmo^{-/-}$ mice were infected with HSV-1 for 8 hours. According to the manufacturer's instructions, these samples were collected for total RNA extraction and then used to generate RNA-seq libraries with a TruSeq PE Cluster Kit v4-cBot-HS (Illumina, USA). The prepared libraries were sequenced on an Illumina platform by Sangon Biotech (Shanghai, China). Genes with P values < 0.05 and fold change > 1.5 were differentially expressed. Gene Ontology (GO) analysis was performed by using the GO knowledgebase (https://geneontology.org/), and the Kyoto Encyclopedia of Genes and Genomes (KEGG) analysis was performed by using the KEGG database (hyyp://www.kegg.jp). The volcano plot was drawn by the Volcano mapping tool in SangerBox, and the GraphPad Prism software drew the heatmap.

## Co-immunoprecipitation

The cells were harvested using NP40 lysis buffer containing protease inhibitors (Beyotime, China). After incubation on ice for 30min, the lysates were centrifuged at 13,000 rpm for 20 min at 4˚C. Subsequently, a proportion of the cell lysate was performed for analysis as input, and another proportion was subjected to precipitation with appropriate antibodies overnight

at 4˚C. The next day, the beads were washed three times with wash buffer (50 mM Tris pH 7.4, 500 mM NaCl, 0.1% (v/v) NP-40, 1 mM EDTA). The precipitated proteins were eluted from beads or gel by the heating sample in SDS loading buffer at 100˚C for 10 min. The precipitates were subjected to Western Blotting Analysis.

## Viral plaque assay

The Vero cells were seeded in a 12-well plate at a density of $5 \times 10^5$ for 12 h, and the virus samples in 10-fold serial dilution were added. After 2 h of incubation at 37˚C in 5% $CO_2$, the supernatant was aspirated, and the cells were washed with PBS. Then, each well added a medium containing 1% FBS, 1% low-melting-point agarose, and 1% penicillin-streptomycin. After 5 days of incubation, the cells were fixed with paraformaldehyde for 1 h and stained with 2% crystal violet for another 1 h. Plaques were visualized and enumerated, and the virus titer was calculated as plaque-forming units per mL (PFU/mL).

## Luciferase assay

The Steady-Glo *Renilla* Luciferase detection system and Dual-Luciferase Reporter Assay were employed according to the manufacturer's instructions (Promega, Madison, WI, USA).

## $Ca^{2+}$ imaging

Calbryte 520 dye is a cell-permeable calcium indicator. Cells were loaded with this dye for 25 min in the dark at room temperature and washed 3 times with PBS. Next, cells were treated with appropriate stimulation. Calbryte 520 fluorescence imaging was recorded under a Leica microscope (Leica, Wetzlar, Germany).

## Cell viability assay

$1 \times 10^4$ cells were inoculated in each well of 96-well-plate, and a series of concentrations of QUIN were added. After incubating for the required time, 10 μl CCK8 solution was added into each well, followed by 1 h in the dark at 37˚C. The OD at 450 nm was then detected by a microplate reader (Biotek, Synergy HTX, USA).

## *In vivo* therapy experiment

C57BL/6 mice were bred in the SPF animal facility of the Laboratory Animal Resource Center of Sun Yat-sen University, and this study was approved by the Institutional Review Boards and Animal Care and Use Committees of Sun Yat-Sen University (Approval No. SYSU-IACUC-2021-000048). 6 to 8-week-old mice were anesthetized, and corneal epithelial debridement was performed using a 30-gauge needle, followed by the inoculation of $10^5$ PFU HSV-1 (McKrae). Intraperitoneal injections of ACV (5 mg/kg), QUIN (2.5 mg/kg), or vehicle alone in 2% DMSO were administered daily for one week, and the ocular swabs, disease scores, and corneal images (Carl Zeiss stereoscope) were acquired during the experiment. The corneal surface was washed with PBS (20 μl/eye) at various times post-infection, and then the virus titer of these samples was quantified by plaque assay.

## Challenge experiment in *kmo*-deficient mice

*kmo*-deficient C57BL/6 mice (*kmo*$^{-/-}$ mice) were constructed and identified as shown in S8 Fig and infected with $3 \times 10^4$ PFU HSV-1 (McKrae) by corneal inoculation as described above. The disease symptoms of experimental mice were monitored, and samples were collected as above to perform the corresponding detection.

### QUIN rescue experiment

Both kmo$^{+/+}$ and kmo$^{-/-}$ C57BL/6 mice were infected by HSV-1 (McKrae) as described above, and then intraperitoneal injections of QUIN (0, 0.5, 1, 2.5 mg/kg) were daily administered for one week. The disease symptoms of experimental mice were monitored, and samples were collected to perform the corresponding detection.

### ELISA assay

The samples were collected and analyzed with the Mouse IFN Beta ELISA Kit (Solarbio Life Science, Beijing, China) according to the manufacturer's instructions.

### ELISPOT

Enzyme-linked immunosorbent spot (ELISPOT) assays were performed as previously described [86]. Briefly, 96-well plates (Millipore, Immobilon-P membrane) were coated with anti-IFN-γ monoclonal antibody (BD Pharmingen) overnight at 4˚Cand then blocked with 10% fetal bovine serum for 2 h at 37˚C. Freshly isolated splenocytes were added at $4 \times 10^5$ cells/well, and the HSV-1 peptides (Genscript, Nanjing, China, listed in S3 Table were immediately added at a final concentration of 2 mg/mL. The cells were incubated for 24 h at 37˚C, and the expression of IFN-γ was then detected using biotinylated polyclonal anti-mouse IFN-γ (BD Pharmingen) and NBT/BCIP reagent (Pierce). Finally, the numbers of spots were quantified using an ELISPOT reader (Bioreader4000, BIOSYS, Germany). The data was reported as spot-forming cells (SFC) per million cells.

### Intracellular Cytokine Staining (ICS)

ICS was processed according to our previous method [87]. Briefly, $1\times10^6$ freshly isolated mouse splenocytes were stimulated with HSV-1 peptides (Genscript, Nanjing, China, listed in S3 Table at a final concentration of 2 mg/mL for 2 h at 37˚C. The splenocytes were then incubated with brefeldin A (BD Pharmingen) for 16 h at 37˚C. After incubation, the cells were collected and stained with anti-CD3-FITC, anti-CD4-BB700, and anti-CD8-PE-Cy7 monoclonal antibodies (BD Biosciences) for 1 h. Then the resuspended cells were permeabilized in FACS Perm/wash buffer for 20 min before staining with anti-IFN-γ-Alexa Fluor 647, anti-TNF-α-PE, and anti-IL-2-BV605 (BD Pharmingen). Samples were tested using the flow cytometer (CytoFLEX S, Beckman, America) instrument with CytExpert software (version 2.4).

### The complete blood counts

Peripheral blood and plasma samples of experimental mice were collected following standard protocols, and the complete blood count was conducted with Mindray veterinary automatic blood cell analyzer BC-2800Vet.

### Hematoxylin and eosin (H&E) staining

The different tissues of experimental mice were immersed completely in 4% paraformaldehyde, gradually dehydrated, embedded in paraffin, and cut into sections. H&E staining was performed according to standard protocol by Wuhan Service Biotechnology CO., LTD.

### LC-MS/MS analysis for Trp-Metabolites

The LC-MS/MS analysis of mouse serum (100 μl) was performed by Shanghai Applied Protein Technology, Shanghai, China.

## Statistical analysis

Statistical analyses were performed using GraphPad Prism software version 8 (GraphPad Software, Inc.). Statistical significance was calculated using Student's two-tailed unpaired t-test or ANOVA with Holm-Sidak's multiple comparisons test. $^{*}P < 0.05$; $^{**}P < 0.01$; $^{***}P < 0.001$.

## Supporting information

**S1 Fig. QUIN has a rescue effect on 293T *kmo*$^{-/-}$ cell lines.** 293T *kmo*$^{-/-}$ cell lines were pre-treated with QUIN for 8 h and then infected with HSV-1 at MOI of 0.25 for 8 h. Then, the expressions of HSV-1 and IFN-β were measured by RT-qPCR. The expression level of mRNA was normalized to the expression of β-actin, and the data from at least triplicates were shown as the mean ± SD. $^{*}P < 0.05$, $^{**}P < 0.01$, $^{***}P < 0.001$.
(TIF)

**S2 Fig. QUIN showed no obvious toxicity to different cell lines.** The different concentrations of QUIN were added into 293T, Vero, and Raw264.7 cells for 72 hours, and then the cell viability was detected with CCK8 assay. The final data are presented as the mean ± SD of at least triplicate experiments.
(TIF)

**S3 Fig. RNA-Seq analysis to reveal the modulation of host antiviral signal pathway during viral infections by QUIN treatment.** (A) Representative of the volcano plots to identify the differential gene expression (DGE) between QUIN-treated cells and non-QUIN-treated cells. Adjust *P*-value < 0.05, fold change |FC| > 1.5. Red dots represent those up-regulated genes. Green dots represent those down-regulated genes. Black dots represent those non-changed genes (Non-DEG). (B) The gene ontology (GO) annotation analysis for the related DEGs involved in QUIN treatment. (C) The enrichment analysis of Kyoto encyclopedia of genes and genomes (KEGG) of the related signaling pathways by QUIN treatment (P-value < 0.05). (D) Heatmap of the 19 selected genes involved in antiviral signaling pathway by QUIN treatment (P-value < 0.05).
(TIF)

**S4 Fig. The expression of NMDAR in different cells.** The Western Blotting analysis confirmed NMDAR protein expression in 293T, 293T *kmo*$^{-/-}$, Hela, A549, BMM, THP-1, Raw 264.7 cell lines. Mouse brain protein lysate as the positive control.
(TIF)

**S5 Fig. KMO and QUIN significantly enhanced IFN-β expression during viral infections.** (A) Cells were transfected with different concentrations of KMO-expressing plasmid for 24 h, followed by viral infections for 8 h at MOI of 1, including HSV-1, VSV, ZIKV, DENV4, Ad5, SARS-CoV-2 pseudovirus, and PR8, and then the expression of IFN-β was quantified by RT-qPCR. (B) Cells were pretreated with QUIN at different concentrations for 8 h, followed by viral infections for 8 h at MOI of 1, including HSV-1, VSV, ZIKV, DENV4, Ad5, SARS-CoV-2 pseudovirus, and PR8, and then the expression of IFN-β was quantified by RT-qPCR. The expression level of mRNA was normalized to the expression of β-actin, and the data from at least triplicates were shown as the mean ± SD. $^{*}P < 0.05$, $^{**}P < 0.01$, $^{***}P < 0.001$.
(TIF)

**S6 Fig. QUIN inhibits viral replication in different cell lines.** A549 cell lines and BMM cell lines were pretreated with QUIN (1mM) for 8 h, followed by infected with different viruses at MOI of 1 for 8 h. The expression of viruses was measured by RT-qPCR. The expression level of mRNA was normalized to the expression of β-actin, and the data from at least triplicates

were shown as the mean ± SD. $^*P < 0.05$, $^{**}P < 0.01$, $^{***}P < 0.001$.
(TIF)

**S7 Fig. KMO and QUIN exerted the antiviral effect independent on IDO1.** (A,B) 293T cell lines were pretreated with IDO1 inhibitor 1-MT (10 μM) for 4 h, followed by HSV-1 infection at MOI of 1 for 8 h. (C,D) Cells were pretreated with 1-MT, and then treated with KMO over-expression or QUIN followed by HSV-1 infection. The expression of HSV-1 and IDO1 were measured by RT-qPCR. The expression level of mRNA was normalized to the expression of β-actin, and the data from at least triplicates were shown as the mean ± SD. $^*P < 0.05$, $^{**}P < 0.01$, $^{***}P < 0.001$.
(TIF)

**S8 Fig. Construction and identification of *kmo*$^{-/-}$ mice in this study.** (A) *kmo* knockout strategy. The position shown by the scissors was the two sites of *kmo* exons targeted by designed CRISPR/Cas9. The red line represents the knocked-out fragment. F1/R1 primer pair and F1/R2 primer pair were the positions of the two primers to identify whether the knockout was successful. (B) The *kmo*$^{-/-}$ mice and wild-type mice were identified by F1/R1 primers. Theoretically, the amplified fragment of *kmo*$^{-/-}$ mice should be 574 bp (base pair) in length, while that of wild-type mice is too long to be amplified. (C) The *kmo*$^{-/-}$ mice and wild-type mice were identified by F1/R2 primers. Theoretically, the amplified fragment of wild-type mice should be 514 bp (base pair) in length, while that of *kmo*$^{-/-}$ mice cannot be amplified because of the lack of R2 sequence. (D) The *kmo*$^{-/-}$ mice and wild-type mice were identified by Western Blotting (top) and RT-qPCR (bottom). The expression level of mRNA was normalized to the expression of β-actin, and the data from at least triplicates were shown as the mean ± SD. $^*P < 0.05$, $^{**}P < 0.01$, $^{***}P < 0.001$.
(TIF)

**S9 Fig. RNA-Seq data to analyze the transcriptome difference of primary macrophages between *kmo*$^{-/-}$ mice and *kmo*$^{+/+}$ mice.** (A) Representative of the volcano plots to identify the differential gene expression (DGE) between bone marrow-derived macrophages from wild *kmo*$^{+/+}$ mice and *kmo*$^{-/-}$ mice. Adjust P-value < 0.05, fold change |FC| > 1.5. Red dots represent those up-regulated genes. Green dots represent those down-regulated genes. Black dots represent those non-changed genes (Non-DEG). (B) The gene ontology (GO) annotation analysis for the upregulated DEGs in bone marrow-derived macrophages from wild *kmo*$^{+/+}$ mice compared to *kmo*$^{-/-}$ mice. (C) The enrichment analysis of Kyoto encyclopedia of genes and genomes (KEGG) of the upregulated signaling pathways in bone marrow-derived macrophages from wild *kmo*$^{+/+}$ mice compared to *kmo*$^{-/-}$ mice (P-value < 0.05). (D) Heatmap of the selected genes involved in antiviral signaling pathway (P-value < 0.05).
(TIF)

**S10 Fig. QUIN has a rescue effect in virus-infected *kmo*$^{-/-}$ mice.** (A) Schedule for evaluating the rescue effect of QUIN on antiviral infections in *kmo*$^{-/-}$ mice. Briefly, both *kmo*$^{+/+}$ C57BL/6 mice and *kmo*$^{-/-}$ C57BL/6 mice were corneally inoculated with $1×10^5$ PFU HSV-1 McKrae strain, disease symptoms of experimental mice were monitored until 7 dpi. (B,C) Survival curve and disease score of experimental mice in different groups over time post-infection (n = 5 per group). (D) The HSV-1 titer in the eye washing fluid at 6 dpi was measured by plaque assay (n = 3 per group). (E) The concentration of IFN-β in the serum of experimental mice at 7 dpi was measured by ELISA assay (n = 3 per group). The final data are presented as the mean ± SD of triplicate experiments. $^*P < 0.05$, $^{**}P < 0.01$, $^{***}P < 0.001$.
(TIF)

**S11 Fig. The kynurenine, a metabolic substrate of KMO, was accumulated in *kmo*^-/-^ mice.** The blood of virus-infected and non-virus-infected *kmo*^+/+^ mice and *kmo*^-/-^ mice were collected by orbital bleeding, and the plasma was separated to determine tryptophan metabolites including Trytophan (A), N-formyl-kynurenine (B), Kynurenine (C). The final data are presented as the mean ± SD. $^*P < 0.05$, $^{**}P < 0.01$, $^{***}P < 0.001$.
(TIF)

**S1 Table. The sequences of siRNA, sgRNA, shRNA used in this study.**
(DOCX)

**S2 Table. Primers for RT-qPCR used in this study.**
(DOCX)

**S3 Table. The sequence of HSV-1 peptide used in this study.**
(DOCX)

**S4 Table. Key Resources used in this study.**
(DOCX)

## Acknowledgments

The authors thank all members of the Sun lab for their comments on the manuscript.

## Author Contributions

**Conceptualization:** Caijun Sun.

**Data curation:** Jin Zhao, Jiaoshan Chen.

**Formal analysis:** Caijun Sun.

**Funding acquisition:** Caijun Sun.

**Investigation:** Jin Zhao, Jiaoshan Chen, Congcong Wang, Yajie Liu, Minchao Li.

**Methodology:** Jin Zhao, Jiaoshan Chen, Congcong Wang, Yajie Liu, Yanjun Li, Ruiting Li.

**Project administration:** Caijun Sun.

**Resources:** Junjian Wang, Ling Chen, Yuelong Shu, Genhong Cheng, Caijun Sun.

**Software:** Jin Zhao, Jiaoshan Chen, Congcong Wang, Yajie Liu, Minchao Li.

**Supervision:** Caijun Sun.

**Validation:** Jiaoshan Chen, Congcong Wang, Yajie Liu.

**Visualization:** Jin Zhao, Jiaoshan Chen.

**Writing – original draft:** Jin Zhao, Zirong Han, Genhong Cheng, Caijun Sun.

**Writing – review & editing:** Jin Zhao, Zirong Han, Genhong Cheng, Caijun Sun.

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
