## [Decision Letter · Decision Letter 0]

7 Dec 2021

Dear Dr. Sun,

Thank you very much for submitting your manuscript "Kynurenine-3-monooxygenase (KMO) broadly inhibits viral infection via triggering NMDAR/Ca2+ influx and CaMKII/ IRF3-mediated IFN-β production" for consideration at PLOS Pathogens. As with all papers reviewed by the journal, your manuscript was reviewed by members of the editorial board and by several independent reviewers. In light of the reviews (below this email), we would like to invite the resubmission of a significantly-revised version that takes into account the reviewers' comments.

We cannot make any decision about publication until we have seen the revised manuscript and your response to the reviewers' comments. Your revised manuscript is also likely to be sent to reviewers for further evaluation.

Sincerely,

Jun Zhao, Ph.D.,

Guest Editor

PLOS Pathogens

Adolfo García-Sastre

Section Editor

PLOS Pathogens

Kasturi Haldar

Editor-in-Chief

PLOS Pathogens

orcid.org/0000-0001-5065-158X

Michael Malim

Editor-in-Chief

PLOS Pathogens

orcid.org/0000-0002-7699-2064

Reviewer's Responses to Questions

**Part I - Summary**

Reviewer #1: Exploring the novel antiviral agents is a public health priority against frequent outbreaks of emerging infectious diseases. In this manuscript, Zhao and colleagues reported that KMO and its enzymatic product QUIN exerted a novel broad-spectrum antiviral function in vitro and in vivo via triggering NMDAR/Ca2+ influx and CaMKII/ IRF3-mediated IFN-� production. Previously, KMO was recognized as a key rate-limiting enzyme in the Tryptophan metabolism, but its role in the innate immunity in response to viral infections has not been clarified. Thus, this work addresses an important topic to help determine the possibility of applying KMO as a novel target of antiviral drugs. The authors' intriguing data is therefore of considerable interest. There are still some issues to be addressed by the authors that would improve the manuscript.

Reviewer #2: This study showed that the mid-stream enzyme from the kynurenine pathway of the tryptophan metabolism, KMO, plays a role in modulating anti-viral activity. They showed that KMO and quinolinic acid (QA) can exert antiviral activity involving NMDA-induced calcium flux and IRF-mediated IFN-beta activity. They further showed that KMO expression and QA were responsible for inhibiting a broad spectrum of viruses that may confer a protective and susceptibility role in the host against viral infection.

A major strength of the study is that the authors used various approaches to examine the role of KMO and QA as novel antiviral targets. For this, I appreciate and acknowledge the efforts of this study. Another interesting result from this study that was understated in this study is the result of KMO and QA having a direct impact on the immune activity in driving antiviral activity.

A major concern I have is to what extent does IDO-1 involved in this study co-commitment with KMO activity? This is something that needs to be teased out since IDO-1 also has antiviral activity. For example, O Adams et al (J Virol, 2004 - https://dx.doi.org/10.1128/JVI.78.5.2632-2636.2004) showed that IDO-1 can exert antiviral effect via interferon activity. We know that viral infection can trigger IDO-1, which is likely to be the case in this study. Did the authors look at IDO-1 expression to make sure their result is due to KMO and not IDO-1? It is also possible that inhibition of KMO leads to accumulation of kynurenine which can have a negative feedback inhibition on IDO-1, thereby suppressing antiviral activity as well. Hence this is important missing info to look into.

In short, if the authors can prove that the antiviral effect is due to KMO and not IDO. This will be a novel role in the field of tryptophan research.

Reviewer #3: This study examined the mechanism of action of kynurenine-3-monooxygenase (KMO)-mediated antiviral effect. It was shown that the enzymatic product of KMO, quinolinic acid (QUIN), activates N-methyl-d-aspartate receptor (NMDAR) and Ca2+ influx, which subsequently activates Calcium/calmodulin29 dependent protein kinase II (CaMKII) / Interferon regulatory factor 3 (IRF3). Over-expression of KMO led to the inhibition of viral replication, while knockout or mutate KMO promote viral replication. Similarly, QUIN treatment led to the inhibition of viral replication of multiple viruses in cell culture. QUIN was also shown to have in vivo antiviral efficacy in HSV-1 infection mouse model.

**Part II – Major Issues: Key Experiments Required for Acceptance**

Reviewer #1: 1. The author analyzed the difference in transcriptomics of cell lines after virus infections with or without QUIN incubation and found the changes in the interferon pathway. It is recommended to analyze further the difference in transcriptomics of primary cells between KMO-/- mice and wild KMO+/+ mice after viral infections.

2. Does QUIN have rescue effects on KMO-/- cell lines or KMO-/- mice infected with the virus?

3. Fig S5 provided some vital information regarding the immune dysfunction status in kmo-/- mice compared to wild mice, and thus this figure should be moved to the Text.

4. Line 269-271 and Line 285-288: repeated descriptions

5. Line 288-290: The cited literature seems not related to the topic what authors meant

6. phrases such as in vitro, in vivo, and KMO (or kmo) is sometimes in italics and sometimes not in italics

Reviewer #2: 1) Demonstrate that all antiviral activities is independent of IDO-1 expression.

2) If IDO-1 expression confound with the finding, does pharmacological inhibition of IDO-1 (e.g. 1-MT) still confer antiviral activity by KMO?

Reviewer #3: 1. The antiviral effect shown in Fig. 4 is very moderate, and in most case less than 1 log viral reduction was observed. It is also not shown whether the antiviral effect is cell type dependent. For example, does QUIN inhibit influenza virus replication in epithelial cells?

2. NMDA receptor is found in neurons. However, not all the viruses studied in this ms including HSV-1, VSV, PR8, ZIKA, Dengue, and SARS-CoV-2 are neurotropic viruses. Therefore there is a lack of physiological relevance for the use of NMDA activator as antivirals. Are the cell lines used for the antiviral assay all express NMDAR? If yes, what are the evidences?

3. The claim of repurposing quinolinic acid as a broad-spectrum antiviral is misleading. Quinolinic acid (QUIN) is a neuroactive metabolite, and can induce oxidative neurotoxicity. High quinolinic acid levels have been associated with Alzheimer's disease, anxiety, depression, epilepsy, and Huntington's disease. Therefore, a section should be added to state the well-known side effects of quinolinic acid so the readers are not biased.

Ref:Davis I & Liu A (2015) What is the tryptophan kynurenine pathway and why is it important to neurotherapeutics? Expert Rev Neurother 15, 719–721.

4. In the in vivo study, QUIN is dosed by IP injection, and it has been shown that QUIN could not pass the BBB, therefore, how could it elicit the antiviral effect? A relatively low dose of 2.5mg/kg of QUIN was used, is this sufficient to achieve the in vivo drug concentration that is needed to suppress viral infection?

5. NMDAR antagonists such as MK-801 have been shown to inhibit Japanese encephalitis virus (JEV)-mediated neuropathogenesis (ref below). In addition, NMDAR antagonists were also reported as VSV and influenza antivirals. Therefore, modulation of NMDAR might be a double-edge sword.

Ref: Journal of Neuroinflammation volume 15, Article number: 238 (2018)

Bioorg Med Chem. 2012 Jan 15;20(2):942-8.

**Part III – Minor Issues: Editorial and Data Presentation Modifications**

Reviewer #1: The language needs professional editing, here are a few examples.

Line 21: replace "through kynurenine pathway" with "through the kynurenine pathway"

Line 43: replace "mechanism" with "mechanisms"

Line 44: replace "in vivo" with italics "in vivo"

Line 83:replace "infection" with "infections"

Line 118: replace "overexpression" with "the overexpression"

Line 271: remove the space after "kmo+/+ mice"

Line 202, 204, 225, 319: replace "viral infection" with "viral infections"

Reviewer #2: The overall narrative of the introduction and discussion that links KMO to cancer and neurodegeneration seems slightly out of context with what the study is trying to demonstrate. Hence, the direct focus of the role of KP in immunology and virology should be discussed more thoroughly. Some earlier work on KP (especially IDO-1) and antiviral is missing such as O Adams et al (as mentioned above), Gaeling L et al (2016) - https://doi.org/10.1111/febs.13966, Fernandez-Pol (2001) - https://pubmed.ncbi.nlm.nih.gov/11911246/. The authors may also want to consider linking some of the KP discoveries performed in COVID-19. See this paper: https://doi.org/10.1016/j.cels.2020.10.003.

Another question that is worth discussing in the paper is that if the host uses KMO-mediated QA to suppress viral activity, is there a long-term consequence where increased QA production chronically can promotes neurodegeneration?

Reviewer #3: (No Response)

PLOS authors have the option to publish the peer review history of their article (what does this mean?). If published, this will include your full peer review and any attached files.

Reviewer #1: **Yes: **Chunfu ZHENG

Reviewer #2: **Yes: **Chai K Lim

Reviewer #3: No
---

## [Decision Letter · Decision Letter 1]

14 Feb 2022

Dear Dr. Sun,

We are pleased to inform you that your manuscript 'Kynurenine-3-monooxygenase (KMO) broadly inhibits viral infections via triggering NMDAR/Ca2+ influx and CaMKII/ IRF3-mediated IFN-β production' has been provisionally accepted for publication in PLOS Pathogens.

Best regards,

Jun Zhao, Ph.D.,

Guest Editor

PLOS Pathogens

Adolfo García-Sastre

Section Editor

PLOS Pathogens

Kasturi Haldar

Editor-in-Chief

PLOS Pathogens

orcid.org/0000-0001-5065-158X

Michael Malim

Editor-in-Chief

PLOS Pathogens

orcid.org/0000-0002-7699-2064

Reviewer Comments (if any, and for reference):

Reviewer's Responses to Questions

**Part I - Summary**

Reviewer #1: They have well addressed all my concerns, now it is acceptable.

Reviewer #2: This is a revised manuscript. Overall, the authors had addressed the questions raised by the reviewers. I'm happy with all the revision made. Hence, my recommendation is accepting the manuscript. Thanks for the great effort and excellent work!

Reviewer #3: Comments from the previous round of review were largely addressed. I therefore recommend acceptance.

**Part II – Major Issues: Key Experiments Required for Acceptance**

Reviewer #1: They have well addressed all my concerns, now it is acceptable.

Reviewer #2: No issue.

Reviewer #3: (No Response)

**Part III – Minor Issues: Editorial and Data Presentation Modifications**

Reviewer #1: They have well addressed all my concerns, now it is acceptable.

Reviewer #2: No issue

Reviewer #3: (No Response)

PLOS authors have the option to publish the peer review history of their article (what does this mean?). If published, this will include your full peer review and any attached files.

Reviewer #1: **Yes: **Chunfu ZHENG

Reviewer #2: **Yes: **Chai K Lim

Reviewer #3: **Yes: **Jun Wang

---

## [Editor Report · Acceptance letter]

24 Feb 2022

Dear Dr. Sun,

We are delighted to inform you that your manuscript, "Kynurenine-3-monooxygenase (KMO) broadly inhibits viral infections via triggering NMDAR/Ca2+ influx and CaMKII/ IRF3-mediated IFN-β production," has been formally accepted for publication in PLOS Pathogens.

Best regards,

Kasturi Haldar

Editor-in-Chief

PLOS Pathogens

orcid.org/0000-0001-5065-158X

Michael Malim

Editor-in-Chief

PLOS Pathogens

orcid.org/0000-0002-7699-2064